# Combating Representation Learning Disparity with Geometric Harmonization

**Zhihan Zhou**[1]  **Jiangchao Yao**[1,2†]  **Feng Hong**[1]  **Ya Zhang**[1,2]  **Bo Han**[3]  **Yanfeng Wang**[1,2†]

[1]Cooperative Medianet Innovation Center, Shanghai Jiao Tong University
[2]Shanghai AI Laboratory    [3]Hong Kong Baptist University
{zhihanzhou, Sunarker, feng.hong, ya_zhang, wangyanfeng}@sjtu.edu.cn
bhanml@comp.hkbu.edu.hk

## Abstract

Self-supervised learning (SSL) as an effective paradigm of representation learning has achieved tremendous success on various curated datasets in diverse scenarios. Nevertheless, when facing the long-tailed distribution in real-world applications, it is still hard for existing methods to capture transferable and robust representation. Conventional SSL methods, pursuing *sample-level uniformity*, easily leads to representation learning disparity where head classes dominate the feature regime but tail classes passively collapse. To address this problem, we propose a novel Geometric Harmonization (GH) method to encourage *category-level uniformity* in representation learning, which is more benign to the minority and almost does not hurt the majority under long-tailed distribution. Specially, GH measures the population statistics of the embedding space on top of self-supervised learning, and then infer an fine-grained instance-wise calibration to constrain the space expansion of head classes and avoid the passive collapse of tail classes. Our proposal does not alter the setting of SSL and can be easily integrated into existing methods in a low-cost manner. Extensive results on a range of benchmark datasets show the effectiveness of GH with high tolerance to the distribution skewness. Our code is available at https://github.com/MediaBrain-SJTU/Geometric-Harmonization.

## 1 Introduction

Recent years have witnessed a great success of self-supervised learning to learn generalizable representation [7, 9, 15, 63]. Such rapid advances mainly benefit from the elegant training on the label-free data, which can be collected in a large volume. However, the real-world natural sources usually exhibit the long-tailed distribution [50], and directly learning representation on them can lead to the distortion issue of the embedding space, namely, the majority dominates the feature regime [75] and the minority collapses [47]. Thus, it becomes urgent to pay attention to representation learning disparity, especially as fairness of machine learning draws increasing attention [28, 41, 68, 77].

Different from the flourishing supervised long-tailed learning [29, 46, 68], self-supervised learning under long-tailed distributions is still under-explored, since there is no labels available for the calibration. Existing explorations to overcome this challenge mainly resort to the possible tailed sample discovery and provide the implicit bias to representation learning. For example, BCL [77] leverages the memorization discrepancy of deep neural networks (DNNs) on unknown head classes and tail classes to drive an instance-wise augmentation. SDCLR [28] contrasts the feature encoder and its pruned counterpart to discover hard examples that mostly covers the samples from tail classes, and efficiently enhance the learning preference towards tailed samples. DnC [59] resorts to a divide-and-conquer methodology to mitigate the data-intrinsic heterogeneity and avoid the representation

---

† The corresponding authors are Jiangchao Yao and Yanfeng Wang.

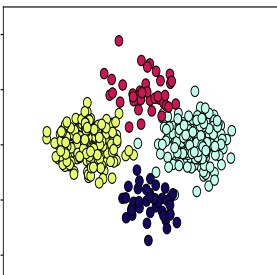 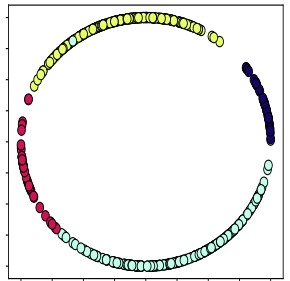 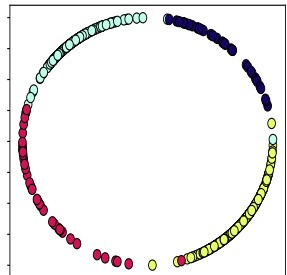

Figure 1: Comparison of Geometric Harmonization and the plain SSL method on a 2-D imbalanced synthetic dataset. (Left) Visualization of the 2-D synthetic dataset. (Middle) The embedding distribution of each category learnt by the vanilla contrastive learning loss is approximately proportional to the number of samples, leading to the undesired representation learning disparity. (Right) GH mitigates the adverse effect of class imbalance and approaches to the category-level uniformity.

collapse of minority classes. Liu et al. [41] adopts a data-dependent sharpness-aware minimization scheme to build support to tailed samples in the optimization. However, few works hitherto have considered the intrinsic limitation of the widely-adopted contrastive learning loss, and design the corresponding balancing mechanism to promote the representation learning parity.

We rethink the characteristic of the contrastive learning loss, and try to understand *"Why the conventional contrastive learning underperforms in self-supervised long-tailed context?"* To answer this question, let us consider two types of representation uniformity: (1) *Sample-level uniformity*. As stated in [62], the contrastive learning targets to distribute the representation of data points uniformly in the embedding space. Then, the feature span of each category is proportional to their corresponding number of samples. (2) *Category-level uniformity*. This uniformity pursues to split the region equally for different categories without considering their corresponding number of samples [49, 19]. In the class-balanced scenarios, the former uniformity naturally implies the latter uniformity, resulting in the equivalent separability for classification. However, in the long-tailed distributions, they are different: sample-level uniformity leads to the feature regime that is biased towards the head classes considering their dominant sample quantity and sacrifices the tail classes due to the limited sample quantity. By contrast, category-level uniformity means the equal allocation *w.r.t.* classes, which balances the space of head and tail classes, and is thus more benign to the downstream classification [17, 19, 37]. Unfortunately, there is no support for promoting category-level uniformity in contrastive learning loss, which explains the question arisen at the beginning.

In this study, we propose a novel method, termed as *Geometric Harmonization* (GH) to combat representation learning disparity in SSL under long-tailed distributions. Specially, GH uses a geometric uniform structure to measure the uniformity of the embedding space in the coarse granularity. Then, a surrogate label allocation is computed to provide a fine-grained instance-wise calibration, which explicitly compresses the greedy representation space expansion of head classes, and constrain the passive representation space collapse of tail classes. The alternation in the conventional loss refers to an extra efficient optimal-transport optimization that dynamically pursues the category-level uniformity. In Figure 1, we give a toy experiment[1] to visualize the distribution of the embedding space without and with GH. In a nutshell, our contributions can be summarized as follows,

1. To our best knowledge, we are the first to investigate the drawback of the contrastive learning loss in self-supervised long-tailed context and point out that the resulted sample-level uniformity is an intrinsic limitation to the representation parity, motivating us to pursue category-level uniformity with more benign downstream generalization (Section 3.2).

2. We develop a novel and efficient *Geometric Harmonization* (Figure 2) to combat the representation learning disparity in SSL, which dynamically harmonizes the embedding space of SSL to approach the category-level uniformity with the theoretical guarantee.

3. Our method can be easily plugged into existing SSL methods for addressing the data imbalance without much extra cost. Extensive experiments on a range of benchmark datasets demonstrate the consistent improvements in learning robust representation with our GH.

---

[1]For more details, please refer to Appendix E.3.

## 2 Related Work

**Self-Supervised Long-tailed Learning.** There are several recent explorations devoted to this direction [40, 28, 59, 41, 77]. BCL [77] leverages the memorization effect of DNNs to automatically drive an instance-wise augmentation, which enhances the learning of tail samples. SDCLR [28] constructs a self-contrast between model and its pruned counterpart to learn more balanced representation. Classic Focal loss [40] leverages the loss statistics to putting more emphasis on the hard examples, which has been applied to self-supervised long-tailed learning [77]. DnC [59] benefits from the parameter isolation of multi-experts during the divide step and the information aggregation during the conquer step to prevent the dominant invasion of majority. Liu et al. [41] proposes to penalize the sharpness surface in a reweighting manner to calibrate class-imbalance learning. Recently, TS [35] employs a dynamic strategy on the temperature factor of contrastive loss, harmonizing instance discrimination and group-wise discrimination. PMSN [2] proposes the power-law distribution prior, replacing the uniform prior, to enhance the quality of learned representations.

**Hyperspherical Uniformity.** The distribution uniformity has been extensively explored from the physic area, *e.g.*, Thomson problem [58, 53], to machine learning area like some kernel-based extensions, *e.g.*, Riesz s-potential [21, 43] or Gaussian potential [12, 4, 62]. Some recent explorations regarding the features of DNNs [49, 17, 47] discover a terminal training stage when the embedding collapses to the geometric means of the classifier *w.r.t.* each category. Specially, these optimal class means specify a perfect geometric uniform structure with clear geometric interpretations and generalization guarantees [78, 67, 31]. In this paper, we first extend this intuition into self-supervised learning and leverage the specific structure to combat the representation disparity in SSL.

## 3 Method

### 3.1 Problem Formulation

We denote the dataset $\mathcal{D}$, for each data point $(\boldsymbol{x}, \boldsymbol{y}) \in \mathcal{D}$, the input $\boldsymbol{x} \in \mathbb{R}^m$ and the associated label $\boldsymbol{y} \in \{1, \ldots, L\}$. Let $N$ denote the number of samples, $\mathrm{R} = N_{max}/N_{min}$ denote the imbalanced ratio (IR), where $N_{max}, N_{min}$ is the number of samples in the largest and smallest class, respectively. Let $n_i$ denote the number of samples in class $i$. In SSL, the ground-truth $\boldsymbol{y}$ can not be accessed and the goal is to transform an image to an embedding via DNNs, *i.e.*, $f_\theta : \mathbb{R}^m \to \mathbb{R}^d$. In the linear probing evaluation, we construct a supervised learning task with balanced datasets. A linear classifier $g(\cdot)$ is built on top of the frozen $f_\theta(\cdot)$ to produce the prediction, *i.e., $g(f_\theta(\boldsymbol{x}))$.*

### 3.2 Geometric Harmonization

As aforementioned, most existing SSL methods in long-tailed context leverage the contrastive learning loss, which encourages the sample-level uniformity in the embedding space. Considering the intrinsic limitation illustrated in Figure 1, we incorporate the geometric clues from the embedding space to calibrate the current loss, enabling our pursuit of category-level uniformity. In the following, we first introduce a specific geometric structure that is critical to Geometric Harmonization.

**Definition 3.1.** (Geometric Uniform Structure). Given a set of vertices $\mathbf{M} \in \mathbb{R}^{d \times K}$, the geometric uniform structure satisfies the following between-class duality

$$\mathbf{M}_i^\top \cdot \mathbf{M}_j = C, \quad \forall i, j \in \{1, 2, \ldots, K\}, \; i \neq j, \tag{1}$$

where $\|\mathbf{M}_i\| = 1, \forall i \in \{1, 2, \ldots, K\}$, $K$ is the number of geometric vertices and $C$ is a constant.

Above structure provides a characteristic that any two vectors in $\mathbf{M}$ have the same angle, namely, the unit space are equally partitioned by the vectors in $\mathbf{M}$. This fortunately follows our expectation about category-level uniformity. Specially, if we use $\mathbf{M}$ as a constant classifier to involve into training, and have the oracle label $\boldsymbol{y}$ of the long-tailed data ($K = L$) to supervise the below prediction

$$\boldsymbol{q} = p(\boldsymbol{y}|f_\theta(\boldsymbol{x}), \mathbf{M}) = \exp\left(\mathbf{M}_y^\top \cdot f_\theta(\boldsymbol{x})/\gamma_{\mathrm{GH}}\right) \Big/ \left(\sum_{i=1}^K \exp\left(\mathbf{M}_i^\top \cdot f_\theta(\boldsymbol{x})/\gamma_{\mathrm{GH}}\right)\right), \tag{2}$$

then according to the neural collapse theory [49], the representation of all samples will fit the geometric uniform structure of $\mathbf{M}$ in the limit, namely, approach category-level uniformity. However,

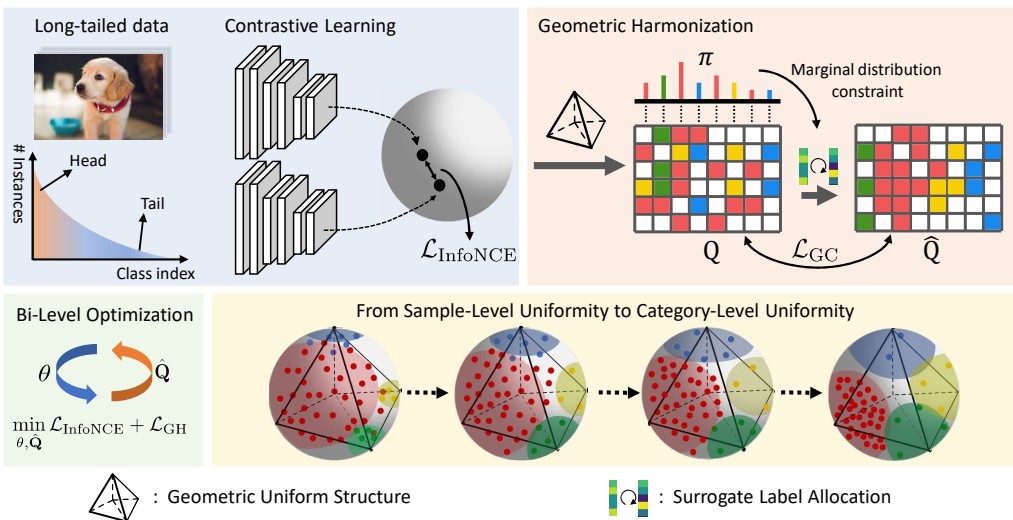

Figure 2: **Overview of Geometric Harmonization** (*w/* InfoNCE). To achieve harmonization with the category-level uniformity, we require ground-truth labels for supervision during training. However, these labels are not available under the self-supervised paradigm. Moreover, estimating surrogate labels directly from the geometric uniform structure is challenging and noisy, especially when the representation is not ideally distributed. To fullfill this gap, we utilize the geometric uniform structure to measure the embedding space, and the captured population statistics are used for an instance-wise calibration by surrogate label allocation, which provides a supervision feedback to counteract the sample-level uniformity. Specially, our method are theoretically grounded to approach category-level uniformity at the loss minimum. The additional model parameters incurred by GH are analytically and empirically demonstrated to be trained in an efficient manner.

the technical challenge is the complete absence of annotation $\boldsymbol{y}$ in our context, making directly constructing the objective $\min \mathbb{E}[\boldsymbol{y} \log \boldsymbol{q}]$ with Eq. (2) impossible to combat the representation disparity.

**Surrogate Label Allocation.** To address the problem of unavailable labels, we explore constructing the surrogate geometric labels $\hat{\boldsymbol{q}}$ to supervise the training of Eq. (2). Concretely, we utilize the recent discriminative clustering idea [1] to acquire such geometric labels, formulated as follows

$$\min_{\hat{\mathbf{Q}}=[\hat{\boldsymbol{q}}_1,\dots,\hat{\boldsymbol{q}}_N]} \mathcal{L}_{\text{GH}} = -\frac{1}{|\mathcal{D}|} \sum_{\boldsymbol{x}_i \sim \mathcal{D}} \hat{\boldsymbol{q}}_i \log \boldsymbol{q}_i, \quad \text{s.t.} \ \hat{\mathbf{Q}} \cdot \mathbb{1}_N = N \cdot \boldsymbol{\pi}, \ \hat{\mathbf{Q}}^\top \cdot \mathbb{1}_K = \mathbb{1}_N, \quad (3)$$

where $\hat{\boldsymbol{q}}_i$ refers to the soft assignment constrained in a $K$-dimensional probability simplex and $\boldsymbol{\pi} \in \mathbb{R}_+^K$ refers to the distribution constraint. As we are dealing with long-tailed data, we propose to use the geometric uniform structure $\mathbf{M}$ to automatically compute the population statistic of the embedding space as $\boldsymbol{\pi}$. Finally, Eq. (3) can be analytically solved by *Sinkhorn-Knopp algorithm* [14] (refer to Appendix D for Algorithm 1). Note that, the above idea builds upon a conjecture: the constructed surrogate geometric labels are mutually correlated with the oracle labels so that they have the similar implication to approach category-level uniformity. We will empirically verify the rationality of such an assumption via *normalized mutual information* in Section 4.4.

**Overall Objective.** Eq. (3) can be easily integrated into previous self-supervised long-tailed learning methods for geometric harmonization, *e.g.,* SDCLR [28] and BCL [77]. For simplicity, we take their conventional InfoNCE loss [48] as an example and write the overall objective as follows

$$\min_{\theta, \hat{\mathbf{Q}}} \mathcal{L} = \mathcal{L}_{\text{InfoNCE}} + w_{\text{GH}} \mathcal{L}_{\text{GH}}, \quad (4)$$

where $w_{\text{GH}}$ represents the weight of the geometric harmonization loss. Optimizing Eq. (4) refers to a bi-level optimization style: in the outer-loop, optimize $\min_{\hat{\mathbf{Q}}} \mathcal{L}_{\text{GH}}$ with fixing $\theta$ to compute the surrogate geometric labels; in the inner-loop, optimize $\min_\theta \mathcal{L}_{\text{InfoNCE}} + \mathcal{L}_{\text{GH}}$ with fixing $\hat{\mathbf{Q}}$ to learn the representation model. The additional cost compared to the vanilla method primarily arises from from the outer-loop, which will be discussed in Section 3.4 and verified in Section 4.4.

Compared with previous explorations [1, 7], the uniqueness of GH lies in the following three aspects: (1) *Geometric Uniform Structure.* The pioneering works mainly resort to a learnable classifier to perform clustering, which can easily be distorted in the long-tailed scenarios [17]. Built on

Table 1: Linear probing of vanilla discriminative clustering methods and variants of GH on CIFAR-100-LT.

| IR | SimCLR | +SeLA | +SwAV | +GH | *w/o* GUS | *w/o* $\pi$ |
|-----|--------|-------|-------|------|-----------|-------------|
| 100 | 50.7   | 50.5  | 52.2  | 54.0 | 53.3      | 53.1        |
| 50  | 52.2   | 52.0  | 53.0  | 55.4 | 54.6      | 54.4        |
| 10  | 55.7   | 56.0  | 56.1  | 57.4 | 56.7      | 56.4        |

the geometric uniform classifier, our method is capable to provide high-quality clustering results with clear geometric interpretations. (2) *Flexible Class Prior.* The class prior $\pi$ is assumed to be uniform among the previous attempts. When moving to the long-tailed case, this assumption will strengthen the undesired sample-level uniformity. In contrast, our methods can potentially cope with any distribution with the automatic surrogate label allocation. (3) *Theoretical Guarantee.* GH is theoretically grounded to achieve the category-level uniformity in the long-tailed scenarios (refer to Section 3.3), which has never been studied in previous methods. To gain more insights into our method, we further compare GH with discriminative clustering methods (SeLA [1], SwAV [7]) and investigate the impact of various components in GH. From the results in Table 1, we can see that GH consistently outperforms the vanilla discriminative clustering baselines in the linear probing evaluation. Notably, we observe that both GUS and the class prior $\pi$ play a critical role in our GH.

### 3.3 Theoretical Understanding

Here, we reveal the theoretical analysis of GH on promoting the representation learning to achieve category-level uniformity instead of sample-level uniformity. Let us begin with a deteriorated case of sample-level uniformity under the extreme imbalance, *i.e.*, minority collapse [17].

**Lemma 3.2.** *(Minority collapse) Assume the samples follow the uniform distribution $n_1 = n_2 = \cdots = n_{L_H} = n_H$, $n_{L_H+1} = n_{L_H+2} = \cdots = n_L = n_T$ in head and tail classes respectively. Assume $d \geq L$ and $n_H/n_T \to +\infty$, the lower bound (Lemma C.1) of $\mathcal{L}_{\text{InfoNCE}}$ achieves the minimum when the class means of the tail classes collapse to an identical vector:*

$$\lim \boldsymbol{\mu}_i - \boldsymbol{\mu}_j = \mathbf{0}_L, \ \forall L_H \leq i \leq j \leq L,$$

*where $\boldsymbol{\mu}_l = \frac{1}{n_l} \sum_{i=1}^{n_l} f_\theta(\boldsymbol{x}_{l,i})$ denotes the class means and $\boldsymbol{x}_{l,i}$ is the i-th data point with label l.*

This phenomenon indicates all representations of the minority will collapse completely to one point without considering the category discrepancy, which aligns with our observation regarding the passive collapse of tailed samples in Figure 1. To further theoretically analyze GH, we first quantitatively define category-level uniformity in the following, and then theoretically claim that with the geometric uniform structure (Definition. 3.1) and the perfect aligned allocation (Eq. (3)), we can achieve the loss minimum at the stage of realizing category-level uniformity.

**Definition 3.3.** *(Categorical-level Uniformity)* We define categorical-level uniformity on the embedding space *w.r.t* the geometric uniform structure $\mathbf{M}$ when it satisfies

$$\boldsymbol{\mu}_k^* = \mathbf{M}_k, \ \forall k = 1, 2, \ldots, K,$$

where $\boldsymbol{\mu}_k^* = \frac{1}{n_k} \sum_{i=1}^{n_k} f_\theta^*(\boldsymbol{x}_{k,i})$ represents the class mean for samples assigned with the surrogate geometric label $k$ in the embedding space.

**Theorem 3.4.** *(Optimal state for $\mathcal{L}$) Given Eq. (4) under the proper optimization strategy, when it arrives at the category-level uniformity (Definition 3.3) defined on the geometric uniform structure $\mathbf{M}$ (Definition 3.1), we will achieve the minimum of the overall loss $\mathcal{L}^*$ as*

$$\mathcal{L}^* = -2 \sum_{l=1}^{L} \pi_l^{\boldsymbol{y}} \log \left( 1 / \left( 1 + (K-1) \exp(C-1) \right) \right) + \log \left( J/L \right), \tag{5}$$

*where $J$ denotes the size of the collection of the negative samples and $\pi^{\boldsymbol{y}}$ refers to the marginal distribution of the latent ground-truth labels $\boldsymbol{y}$.*

This guarantees the desirable solution with the minimal intra-class covariance and the maximal inter-class covariance under the geometric uniform structure [49, 17], which benefits the downstream generalization. Notably, no matter the data distribution is balanced or not, our method can persistently maintain the theoretical merits on calibrating the class means to achieve category-level uniformity. We also empirically demonstrate the comparable performance with GH on the balanced datasets in Section 4.4, as in this case category-level uniformity is equivalent to sample-level uniformity.

### 3.4 Implementation and Complexity analysis

In Algorithm 2 of Appendix D, we give the complete implementation of our method. One point that needs to be clarified is that we learn the label allocation $\hat{q}$ in the mini-batch manner. In addition, the geometric prediction $q$ and the adjusted $\hat{q}$ are computed at the beginning of every epoch as the population-level statistic will not change much in a few mini-batches. Besides, we maintain a momentum update mechanism to track the prediction of each sample to stabilize the training, *i.e.*, $q^m \leftarrow \beta q^m + (1 - \beta)q$. When combined with the joint-embedding loss, we naturally adopt a cross-supervision mechanism $\min \mathbb{E}[\hat{q}^+ \log q]$ for the reconciliation with contrastive baselines. The proposed method is illustrated in Figure 2 for visualization.

For complexity, assume that the standard optimization of deep neural networks requires forward and backward step in each mini-batch update with the time complexity as $\mathcal{O}(B\Lambda)$, where $B$ is the mini-batch size and $\Lambda$ is the parameter size. At the parameter level, we add an geometric uniform structure with the complexity as $\mathcal{O}(BKd)$, where $K$ is the number of geometric labels and $d$ is the embedding dimension. For Sinkhorn-Knopp algorithm, it only refers to a simple matrix-vector multiplication as shown in Algorithm 1, whose complexity is $\mathcal{O}(E_s(B + K + BK))$ with the iteration step $E_s$. The complexity incurred in the momentum update is $\mathcal{O}(BK)$. Since $K, d$ and $E_s$ are significantly smaller than the model parameter $\Lambda$ of a million scale, the computational overhead involved in GH is negligible compared to $\mathcal{O}(B\Lambda)$. The additional storage for a mini-batch of samples is the matrix $\mathbf{Q}^m \in \mathbb{R}^{K \times B}$, which is also negligible to the total memory usage. To the end, GH incurs only a small computation or memory cost and thus can be plugged to previous methods in a low-cost manner. The empirical comparison about the computational cost is summarized in Table 17.

## 4 Experiments

### 4.1 Experimental Setup

**Baselines.** We mainly choose five baseline methods, including (1) *plain contrastive learning*: Sim-CLR [9], (2) *hard example mining*: Focal [40], (3) *asymmetric network pruning*: SDCLR [28], (4) *multi-expert ensembling*: DnC [59], (5) *memorization-guided augmentation*: BCL [77]. Empirical comparisons with more baseline methods can be referred to Appendix F.3.

**Implementation Details.** Following previous works [28, 77], we use ResNet-18 [23] as the backbone for small-scale dataset (CIFAR-100-LT [5]) and ResNet-50 [23] for large-scale datasets (ImageNet-LT [44], Places-LT [44]). For experiments on CIFAR-100-LT, we train model with the SGD optimizer, batch size 512, momentum 0.9 and weight decay factor $5 \times 10^{-4}$ for 1000 epochs. For experiments on ImageNet-LT and Places-LT, we only train for 500 epochs with the batch size 256 and weight decay factor $1 \times 10^{-4}$. For learning rate schedule, we use the cosine annealing decay with the learning rate $0.5 \to 1e^{-6}$ for all the baseline methods. As GH is combined with baselines, a proper warming-up of 500 epochs on CIFAR-100-LT and 400 epochs on ImageNet-LT and Places-LT are applied. The cosine decay is set as $0.5 \to 0.3$, $0.3 \to 1e^{-6}$ respectively. For hyper-parameters of GH, we provide a default setup across all the experiments: set the geometric dimension $K$ as 100, $w_{\text{GH}}$ as 1 and the temperature $\gamma_{\text{GH}}$ as 0.1. In the surrogate label allocation, we set the regularization coefficient $\lambda$ as 20 and Sinkhorn iterations $E_s$ as 300. Please refer to Appendix E.3 for more experimental details.

**Evaluation Metrics.** Following [28, 77], *linear probing* on a balanced dataset is used for evaluation. We conduct full-shot evaluation on CIFAR-100-LT and few-shot evaluation on ImageNet-LT and Places-LT. For comprehensive performance comparison, we present the linear probing performance and the standard deviation among three disjoint groups, *i.e.*, [many, medium, few] partitions [44].

### 4.2 Linear Probing Evaluation

**CIFAR-100-LT.** In Table 2, we summarize the linear probing performance of baseline methods *w/* and *w/o* GH on a range of benchmark datasets, and provide the analysis as follows.

(1) *Overall Performance.* GH achieves the competitive results *w.r.t* the [many, medium, few] groups, yielding a overall performance improvements averaging as 2.32%, 2.49% and 1.52% on CIFAR-100-LT with different imbalanced ratios. It is worth noting that on the basis of the previous state-of-the-art BCL, our GH further achieves improvements by 1.20%, 1.82% and 1.22%, respectively. Our

Table 2: Linear probing results on CIFAR-100-LT with different imbalanced ratios (100, 50, 10), ImageNet-LT and Places-LT. Many/Med/Few (↑) indicate the average accuracy (%) *w.r.t* fine-grained partitions according to the class cardinality. Std (↓) means the standard deviation of the group-level performance and Avg (↑) is the average accuracy (%) of the full test set. Following the previous work [28, 77], Std represents a balancedness measure to quatify the variance among three specified groups. Improv. (↑) represents the averaging performance improvements *w.r.t.* different baseline methods. **We report the error bars with the multi-run experiments in Table 12 in Appendix F.**

| Dataset | | SimCLR | +GH | Focal | +GH | SDCLR | +GH | DnC | +GH | BCL | +GH | Improv. |
|---|---|---|---|---|---|---|---|---|---|---|---|---|
| CIFAR-R100 | Many | 54.97 | 57.38 | 54.24 | 57.01 | 57.32 | 57.44 | 55.41 | 57.56 | 59.15 | 59.50 | **+1.56** |
| | Med | 49.39 | 52.27 | 49.58 | 52.93 | 50.70 | 52.85 | 51.30 | 53.74 | 54.82 | 55.73 | **+2.35** |
| | Few | 47.67 | 52.12 | 49.21 | 51.74 | 50.45 | 54.06 | 50.76 | 53.26 | 55.30 | 57.67 | **+3.09** |
| | Std | 3.82 | 2.99 | 2.80 | 2.76 | 3.90 | 2.38 | 2.54 | 2.36 | 2.37 | 1.89 | **-0.61** |
| | Avg | 50.72 | 53.96 | 51.04 | 53.92 | 52.87 | 54.81 | 52.52 | 54.88 | 56.45 | 57.65 | **+2.32** |
| CIFAR-R50 | Many | 56.00 | 58.88 | 55.40 | 57.97 | 57.50 | 58.47 | 56.03 | 59.04 | 59.44 | 60.82 | **+2.16** |
| | Med | 50.48 | 53.00 | 51.14 | 53.55 | 51.85 | 53.88 | 52.68 | 55.05 | 54.73 | 57.58 | **+2.44** |
| | Few | 50.12 | 54.27 | 50.02 | 53.58 | 52.15 | 53.58 | 50.83 | 54.81 | 57.30 | 58.55 | **+2.87** |
| | Std | 3.30 | 3.09 | 2.84 | 2.54 | 3.18 | 2.74 | 2.64 | 2.38 | 2.36 | 1.66 | **-0.38** |
| | Avg | 52.24 | 55.42 | 52.22 | 55.06 | 53.87 | 55.34 | 53.21 | 56.33 | 57.18 | 59.00 | **+2.49** |
| CIFAR-R10 | Many | 57.85 | 59.26 | 58.18 | 60.06 | 58.47 | 59.21 | 59.82 | 61.09 | 60.41 | 61.41 | **+1.26** |
| | Med | 55.06 | 56.91 | 55.82 | 56.79 | 54.79 | 56.06 | 56.67 | 58.33 | 57.15 | 59.27 | **+1.57** |
| | Few | 54.03 | 55.85 | 54.64 | 57.24 | 52.97 | 55.58 | 56.21 | 57.33 | 59.76 | 60.30 | **+1.74** |
| | Std | 1.98 | 1.75 | 1.80 | 1.77 | 2.80 | 1.97 | 1.96 | 1.95 | 1.73 | 1.07 | **-0.35** |
| | Avg | 55.67 | 57.36 | 56.23 | 58.05 | 55.44 | 56.97 | 57.59 | 58.94 | 59.12 | 60.34 | **+1.52** |
| ImageNet-LT | Many | 41.69 | 41.53 | 42.04 | 42.55 | 40.87 | 41.92 | 41.70 | 42.19 | 42.92 | 43.22 | **+0.44** |
| | Med | 33.96 | 36.35 | 35.02 | 36.75 | 33.71 | 36.53 | 34.68 | 36.63 | 35.89 | 38.16 | **+2.23** |
| | Few | 31.82 | 35.84 | 33.32 | 36.28 | 32.07 | 36.04 | 33.58 | 35.86 | 33.93 | 36.96 | **+3.25** |
| | Std | 5.19 | 3.15 | 4.62 | 3.49 | 4.68 | 3.26 | 4.41 | 3.45 | 4.73 | 3.32 | **-1.39** |
| | Avg | 36.65 | 38.28 | 37.49 | 38.92 | 36.25 | 38.53 | 37.23 | 38.67 | 38.33 | 39.95 | **+1.68** |
| Places-LT | Many | 31.98 | 32.46 | 31.69 | 32.40 | 32.17 | 32.78 | 32.07 | 32.51 | 32.69 | 33.22 | **+0.55** |
| | Med | 34.05 | 35.03 | 34.33 | 35.14 | 34.71 | 35.60 | 34.51 | 35.55 | 35.37 | 36.00 | **+0.87** |
| | Few | 35.63 | 36.14 | 35.73 | 36.49 | 35.69 | 36.18 | 35.84 | 35.91 | 37.18 | 37.62 | **+0.45** |
| | Std | 1.83 | 1.89 | 2.05 | 2.08 | 1.82 | 1.82 | 1.91 | 1.87 | 2.26 | 2.23 | 0.00 |
| | Avg | 33.61 | 34.33 | 33.65 | 34.42 | 33.99 | 34.70 | 33.90 | 34.52 | 34.76 | 35.32 | **+0.68** |

GH consistently improves performance across datasets with varying degrees of class imbalance, demonstrating its potential to generalize to practical scenarios with more complex distributions. Specially, our method does not require any prior knowledge or assumptions about the underlying data distribution, highlighting the robustness and versatility to automatically adapt to the data.

(2) *Representation Balancedness.* In Section 1, we claim that GH helps compress the expansion of head classes and avoid the passive collapse of tail classes, yielding the more balanced representation distribution. To justify this aspect, we compare the variance in linear probing performance among many/medium/few groups, namely, their groupwise standard deviation. According to Table 2, our GH provides [1.56%, 2.35%, 3.09%], [2.16%, 2.44%, 2.87%] and [1.26%, 1.57%, 1.74%] improvements *w.r.t* [many, medium, few] groups on CIFAR-100-LT-R100/R50/R10 with more preference to the minority classes for representation balancedness. Overall, GH substantially improves the standard deviation by [0.61, 0.38, 0.35] on different levels of imbalance.

**ImageNet-LT and Places-LT.** Table 2 shows the comparison of different baseline methods on large-scale dataset ImageNet-LT and Places-LT, in which we have consistent observations. As can be seen, on more challenging real-world dataset, GH still outperforms other methods in terms of overall accuracy, averaging as 1.68%, 0.68% on ImageNet-LT and Places-LT. Specifically, our method provides [0.44%, 2.23%, 3.25%] and [0.55%, 0.87%, 0.45%] improvements in linear probing *w.r.t.* [many, medium, few] groups on ImageNet-LT and Places-LT. The consistent performance overhead indicates the robustness of our method to deal with long-tailed distribution with different characteristics. Moreover, the averaging improvement of standard deviation is 1.39 on ImageNet-LT, indicating the comprehensive merits of our GH on the minority classes towards the representation balancedness. However, an interesting phenomenon is that the fine-grained performance exhibits a different trend on Places-LT. As can be seen, the performance of head classes is even worse than that of tail classes. The lower performance of the head partition can be attributed to the fact that it contains more chal-

Table 3: Supervised long-tailed learning by finetuning on CIFAR-100-LT, ImageNet-LT and Places-LT. We compare the performance of five self-supervised learning methods as the pre-training stage for downstream supervised logit adjustment [46] method. Improv. (↑) represents the averaging performance improvements *w.r.t.* different baseline methods. Besides, the performance of logit adjustment via learning from scratch is also reported for comparisons.

| Dataset | LA | Logit adjustment pretrained with the following SSL methods | | | | | | | | | | Improv. |
|---|---|---|---|---|---|---|---|---|---|---|---|---|
| | | SimCLR | +GH | Focal | +GH | SDCLR | +GH | DnC | +GH | BCL | +GH | |
| CIFAR-LT | 46.61 | 49.81 | 50.84 | 49.83 | 51.04 | 49.79 | 50.73 | 49.97 | 50.84 | 50.38 | 51.32 | **+1.00** |
| ImageNet-LT | 48.27 | 51.10 | 51.67 | 51.15 | 51.82 | 50.94 | 51.64 | 51.31 | 51.88 | 51.43 | 52.06 | **+0.63** |
| Places-LT | 27.07 | 32.63 | 33.86 | 32.69 | 33.75 | 32.55 | 34.03 | 32.98 | 34.09 | 33.15 | 34.48 | **+1.24** |

Table 4: Image classification on ImageNet, Places and fine-grained visual classification on various fine-grained datasets, pretrained on large-scale long-tailed CC3M and then finetuned.

| | Image Classification | | Fine-Grained Visual Classification | | | | | |
|---|---|---|---|---|---|---|---|---|
| | ImageNet | Places | CUB200 | Aircraft | StanfordCars | StanfordDogs | NABirds | Average |
| SimCLR | 52.06 | 37.65 | 44.61 | 65.89 | 57.63 | 50.99 | 46.86 | 53.20 |
| **+GH** | **53.39** | **38.47** | **45.76** | **68.08** | **60.24** | **52.88** | **47.58** | **54.91** |

lenging classes. As a result, we observe that the standard deviation of our GH does not significantly decrease on Places-LT, which requires more effort and exploration for improvement alongside GH.

## 4.3 Downstream Finetuning Evaluation

**Downstream supervised long-tailed learning.** Self-supervised learning has been proved to be beneficial as a pre-training stage of supervised long-tailed recognition to exclude the explicit bias from the class imbalance [68, 41, 77].

To validate the effectiveness of our GH, we conduct self-supervised pre-training as the initialization for downstream supervised classification tasks on CIFAR-100-LT-R100, ImageNet-LT and Places-LT. The state-of-the-art logit adjustment [46] is chosen as the downstream baseline. The combination of GH + LA can be interpreted as a compounded method where GH aims at the re-balanced representation extraction and LA targets the classifier debiasing. In Table 3, we can find that the superior performance improvements are achieved by self-supervised pre-training over the plain supervised learning baseline. Besides, our method can also consistently outperform other SSL baselines, averaging as 1.00%, 0.63% and 1.24% on CIFAR-100-LT-R100, ImageNet-LT and Places-LT. These results demonstrate that GH are well designed to facilitate long-tailed representation learning and improve the generalization for downstream supervised tasks.

**Cross-dataset transfer learning**. To further demonstrate the representation transferability of our GH, we conduct more comprehensive experiments on the large-scale, long-tailed dataset CC3M [52] with various cross-dataset transferring tasks, including downstream classification, object detection and instance segmentation. Specifically, we report the finetuning classification performance on ImageNet, Places and fine-grained visual datasets Caltech-UCSD Birds (CUB200) [61], Aircrafts [45], Stanford Cars [33], Stanford Dogs [32], NABirds [60]. Besides, we evaluate the quality of the learned representation by finetuning the model for object detection and instance segmentation on COCO2017 benchmark [39]. As shown in Tables 4 and 5, we can see that our proposed GH consistently outperforms the baseline across various tasks and datasets. It further demonstrates the importance of considering long-tailed data distribution under large-scale unlabeled data in the pre-training stage. This can potentially be attributed to that our geometric harmonization motivates a more balanced and general emebdding space, improving the generalization ability of the pretrained model to a range of real-world downstream tasks.

## 4.4 Further Analysis and Ablation Studies

**Dimension of Geometric Uniform Structure.** As there is even no category number $L$ available in SSL paradigm, we empirically compare our GH with different geometric dimension $K$ on CIFAR-100-LT-R100, as shown in Figure 3(a). From the results, GH is generally robust to the change of

Table 5: Object detection and instance segmentation with finetuned features on COCO2017 benchmark, pretrained on large-scale long-tailed CC3M.

| | Object Detection | | | Instance Segmentation | | |
|---|---|---|---|---|---|---|
| | $AP^{bbox}$ | $AP^{bbox}_{50}$ | $AP^{bbox}_{75}$ | $AP^{mask}$ | $AP^{mask}_{50}$ | $AP^{mask}_{75}$ |
| SimCLR | 31.7 | 51.0 | 33.9 | 30.2 | 49.8 | 32.1 |
| +GH | **32.7** | **52.2** | **35.2** | **31.1** | **50.8** | **33.0** |

Table 6: Inter-class uniformity (↑) and neighborhood uniformity (↑) of pretrained features on CIFAR-LT.

| | Inter-class Uniformity | | Neighborhood Uniformity | |
|---|---|---|---|---|
| | SimCLR | +GH | SimCLR | +GH |
| C100 | 1.00 | **2.80** | 0.72 | **2.00** |
| C50 | 1.23 | **2.73** | 0.91 | **1.94** |
| C10 | 1.18 | **2.60** | 0.85 | **1.83** |

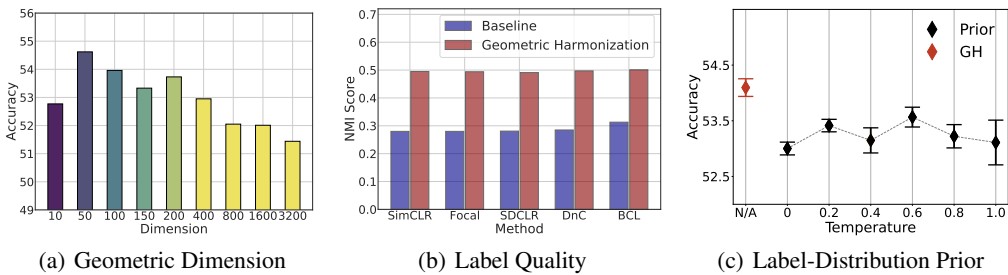

| (a) Geometric Dimension | (b) Label Quality | (c) Label-Distribution Prior |
|---|---|---|

Figure 3: (a) Linear probing performance *w.r.t.* the dimension $K$ of the geometric uniform structure **M** (Appendix B) on CIFAR-LT-R100. (b) NMI score between the surrogate geometric labels and the ground-truth labels in the training stage on CIFAR-LT-R100. (c) Average linear probing and the error bars of the surrogate label allocation with variants of the label prior on CIFAR-LT-R100.

$K$, but slightly exhibits the performance degeneration when the dimension is extremely large or small. Intuitively, when $K$ is extremely large, our GH might pay more attention to the uniformity among sub-classes, while the desired uniformity on classes is not well guaranteed. Conversely, when $K$ is extremely small, the calibration induced by GH is too coarse that cannot sufficiently avoid the internal collapse within each super-class. For discussions of the structure, it can refer to Appendix B.

**Surrogate Label Quality Uncovered.** To justify the effectiveness of surrogate label allocation, we compare the NMI scores [54] between the surrogate and ground-truth label in Figure 3(b). We observe that GH significantly improves the NMI scores across baselines, indicating that the geometric labels are effectively calibrated to better capture the latent semantic information. Notably, the improvements of the existing works are marginal, which further verifies the superiority of GH.

**Exploration with Other Label-Distribution Prior.** To further understand $\boldsymbol{\pi}$, we assume the ground-truth label distribution is available and incorporate the oracle $\boldsymbol{\pi}^{\boldsymbol{y}}$ into the surrogate label allocation. Comparing the results of the softened variants $\boldsymbol{\pi}^{\boldsymbol{y}}_{\gamma_{\mathrm{T}}}$ with the temperature $\gamma_{\mathrm{T}}$ in Figure 3(c), we observe that GH outperforms all the counterparts equipped with the oracle prior. A possible reason is that our method automatically captures the inherent geometric statistics from the embedding space, which is more reconcilable to the self-supervised learning objectives.

**Uniformity Analysis.** In this part, we conduct experiments with two uniformity metrics [62, 37]:

$$\mathrm{U} = \frac{1}{L(L-1)} \sum_{i=1}^{L} \sum_{j=1, j \neq i}^{L} ||\boldsymbol{\mu}_i - \boldsymbol{\mu}_j||_2, \quad \mathrm{U}_k = \frac{1}{Lk} \sum_{i=1}^{L} \min_{j_1, \dots, j_k} \left( \sum_{m=1}^{k} ||\boldsymbol{\mu}_i - \boldsymbol{\mu}_{j_m}||_2 \right),$$

where $j_1, \dots, j_k \neq i$ represent different classes. Specifically, U evaluates average distances between different class centers and $\mathrm{U}_k$ measures how close one class is to its neighbors. As shown in Table 6, our GH outperforms in both inter-class uniformity and neighborhood uniformity when compared with the baseline SimCLR [9]. This indicates that vanilla contrastive learning struggles to achieve the uniform partitioning of the embedding space, while our GH effectively mitigates this issue.

**Comparison with More SSL Methods.** In Table 7, we present a more comprehensive comparison of different SSL baseline methods, including MoCo-v2 [24], MoCo-v3 [11] and various non-contrastive methods such as SimSiam [10], BYOL [20] and Barlow Twins [71]. From the results, we can see that the combinations of different SSL methods and our GH can achieve consistent performance improvements, averaging as 2.33%, 3.18% and 2.21% on CIFAR-100-LT. This demonstrates the prevalence of representation learning disparity under data imbalance in general SSL settings.

Table 7: Linear probing of more SSL variants on CIFAR-100-LT with different imbalanced ratios.

| Dataset | SimSiam | +GH | Barlow | +GH | BYOL | +GH | MoCo-v2 | +GH | MoCo-v3 | +GH |
|---------|---------|-----|--------|-----|------|-----|---------|-----|---------|-----|
| CIFAR-R100 | 49.01 | 51.43 | 48.70 | 51.23 | 51.43 | 52.87 | 51.49 | 53.53 | 54.08 | 55.82 |
| CIFAR-R50 | 48.98 | 53.54 | 49.29 | 51.95 | 52.04 | 53.84 | 52.68 | 55.01 | 55.34 | 56.45 |
| CIFAR-R10 | 55.51 | 57.03 | 53.11 | 56.34 | 55.86 | 57.28 | 58.23 | 60.11 | 59.10 | 60.57 |

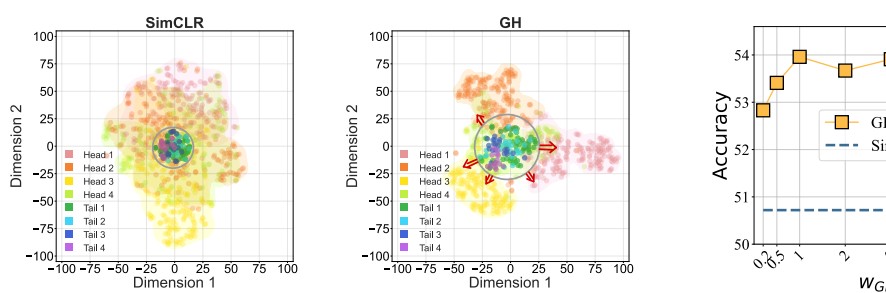

Figure 4: T-SNE visualization of the pretrained features from 8 randomly selected classes *w.r.t* CIFAR-LT training images.

Figure 5: Ablations of $w_{GH}$ on CIFAR-LT.

**On Importance of Bi-Level Optimization.** In Table 8, we empirically compare the direct joint optimization strategy to Eq. (4). From the results, we can see that the joint optimization (*w/* or *w/o* the warm-up strategy) does not bring significant performance improvement over SimCLR compared with that of our bi-level optimization, probably due to the undesired collapse in label allocation [1]. This demonstrates the necessity of the proposed bi-level optimization for Eq. (4) to stabilize the training.

Table 8: Linear probing of joint optimization on CIFAR-100-LT with different IRs.

| Method | C100 | C50 | C10 |
|--------|------|-----|-----|
| SimCLR | 50.72 | 52.24 | 55.67 |
| +GH (Joint) | 50.18 | 52.31 | 54.98 |
| w/ warm-up | 51.14 | 52.75 | 55.37 |
| +GH (Bi-level) | 53.96 | 55.42 | 57.36 |

**Qualitative Visualization.** We conduct t-SNE visualization of the learned features to provide further qualitative intuitions. For simplity, we randomly selected four head classes and four tail classes on CIFAR-LT to generate the t-SNE plots. Based on the results in Figure 4, the observations are as follows: (1) SimCLR: head classes exhibit a large presence in the embedding space and heavily squeeze the tail classes, (2) GH: head classes reduce their occupancy, allowing the tail classes to have more space. This further indicates that the constructed surrogate labels can serve as the high-quality supervision, effectively guiding the harmonization towards the geometric uniform structure.

**Sensitivity Analysis.** To further validate the stability of our GH, We conduct empirical comparison with different weight $w_{GH}$, temperature $\gamma_{GH}$, regularization coefficient $\lambda$ and Sinkhorn iteration $E_s$ on CIFAR-LT, as shown in Figures 5 and 7. From the results, we can see that our GH can consistently achieve satisfying performance with different hyper-parameter.

## 5 Conclusion

In this paper, we delve into the defects of the conventional contrastive learning in self-supervised long-tail context, *i.e.*, representation learning disparity, motivating our exploration on the inherent intuition for approaching the category-level uniformity. From the geometric perspective, we propose a novel and efficient Geometric Harmonization algorithm to counteract the long-tailed effect on the embedding space, *i.e*, over expansion of the majority class with the passive collapse of the minority class. Specially, our proposed GH leverages the geometric uniform structure as an optimal indicator and manipulate a fine-grained label allocation to rectify the distorted embedding space. We theoretically show that our proposed method can harmonize the desired geometric property in the limit of loss minimum. It is also worth noting that our method is orthogonal to existing self-supervised long-tailed methods and can be easily plugged into these methods in a lightweight manner. Extensive experiments demonstrate the consistent efficacy and robustness of our proposed GH. We believe that the geometric perspective has the great potential to evolve the general self-supervised learning paradigm, especially when coping with the class-imbalanced scenarios.

## Acknowledgement

This work was supported by the National Key R&D Program of China (No. 2022ZD0160702), STCSM (No. 22511106101, No. 22511105700, No. 21DZ1100100), 111 plan (No. BP0719010) and National Natural Science Foundation of China (No. 62306178). BH was supported by the NSFC Young Scientists Fund No. 62006202, NSFC General Program No. 62376235, and Guangdong Basic and Applied Basic Research Foundation No. 2022A1515011652.

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
