# Appendix: Combating Representation Learning Disparity with Geometric Harmonization

## Contents

## Reproducibility Statement

We provide our source codes to ensure the reproducibility of our experimental results. Below we summarize several critical aspects *w.r.t* the reproducible results:

- **Datasets.** The datasets we used are all publicly accessible, which is introduced in Appendix E.1. For long-tailed subsets, we strictly follows previous work [29] on CIFAR-100-LT to avoid the bias attribute to the sampling randomness. On ImageNet-LT and Places-LT, we employ the widely-used data split first introduced in [44].

- **Source code.** Our code is available at https://github.com/MediaBrain-SJTU/Geometric-Harmonization.

- **Environment.** All the experiments are conducted on NVIDIA GeForce RTX 3090 with Python 3.7 and Pytorch 1.7.

## A    Additional Discussions of Related Works

### A.1    Supervised Long-tailed Learning

As the explorations on the classifier learning [29, 70] are orthogonal to the self-supervised learning paradigms, we mainly focus on the representation learning in supervised long-tailed recognition. The pioneering work [29] first explores representation and classifier learning with a disentangling mechanisms and shows the merits of instance-balanced sampling strategy on the representation learning stage. Subsequently, Yang and Xu [68] points out the negative impact of label information and proposes to improve the representation learning with semi-supervised learning and self-supervised learning. This motivates a stream of research works diving into the representation learning. Supervised contrastive learning [30, 13] is leveraged with rebalanced sampling or prototypical learning design to pursue a more balanced representation space. Li et al. [37] explicitly regularizes the class centers to a maximum separation structure with similar drives to the balanced feature space.

## A.2 Contrastive Learning is Still Vulnerable to Long-tailed Distribution

The prior works [30, 41] point out that contrastive learning can extract more balanced features compared with the supervised learning paradigm. However, several subsequent works [28, 77] empirically observes that contrastive learning is still vulnerable to the long-tailed distribution, which motivates their model-pruning strategy [28] and memorization-oriented augmentation [77] to rebalance the representation learning. In this paper, we delve into the intrinsic limitation of the contrastive learning method in the long-tailed context, *i.e*, approaching sample-level uniformity to deteriorate the embedding space.

## A.3 Unsupervised Clustering

Deep Cluster [6] applies K-Means clustering to generate pseudo-labels for the unlabeled data, which are then iteratively leveraged as the supervised signal to train a classifier. SeLa [1] first casts the pseudo-label generation as an optimal transport problem and leverages a uniform prior to guide the clustering. SwAV [7] adopts mini-batch clustering instead of dataset-level clustering, enhancing the practical applicability of the optimal transport-based clustering method. Subsequently, Li et al. [36] combines clustering and contrastive learning objectives in an Expectation-Maximization framework, recursively updating the data features towards their corresponding class prototypes. In this paper, we propose a novel Geometric Harmonization method that is capable to cope with long-tailed distribution, the uniqueness can be summarized in the following aspects: (1) *Geometric Uniform Structure*. The pioneering works [1, 7] mainly resort to a learnable classifier to perform clustering, which can easily be distorted in the long-tailed scenarios [17]. Built on the geometric uniform structure, our method is capable to provide high-quality clustering results with clear geometric interpretations. (2) *Flexible Class Prior*. The class prior in [1, 7] is assumed to be uniform among the previous attempts. When moving to the long-tailed case, this assumption will strengthen the undesired sample-level uniformity. In contrast, our methods can potentially cope with any distribution with the automatic surrogate label allocation. (3) *Theoretical Guarantee*. GH is theoretically grounded to achieve the category-level uniformity in the long-tailed scenarios, which has never been studied in previous methods.

## A.4 Taxonomy of Self-supervised Long-tailed Methods

We summarize the detailed taxonomy of self-supervised long-tailed methods in Algorithm 1.

Table 9: Taxonomy of self-supervised long-tailed methods.

| Method | Aspect | Description |
|--------|--------|-------------|
| Focal [40] | Sample Reweighting | Hard example mining |
| rwSAM [41] | Optimization Surface | Data-dependent sharpness-aware minimization |
| SDCLR [28] | Model Pruning | Model pruning and self-contrast |
| DnC [59] | Model Capacity | Multi-expert ensemble |
| BCL [77] | Data Augmentation | Memorization-guided augmentation |
| GH | Loss Limitation | Geometric harmonization |

# B  Discussions of Geometric Uniform Structure (Definition 3.1)

## B.1  Simplex Equiangular Tight Frame ($K \leq d$)

Neural collapse [47] describes a phenomenon that with the training, the geometric centroid of representation progressively collapses to the optimal classifier parameter *w.r.t.* each category. The collection of these points builds a special geometric structure, termed as Simplex Equiangular Tight Frame (ETF). Some study that shares the similar spirit is also explored regarding the maximum separation structure [31]. We present its formal definition as follows.

**Definition B.1.** A Simplex ETF is a collection of points in $\mathbb{R}^d$ specified by the columns of the matrix:

$$\mathbf{M}^{\mathrm{ETF}} = \sqrt{\frac{K}{K-1}} \mathbf{U}(\mathbf{I}_K - \frac{1}{K} \mathbb{1}_K \mathbb{1}_K^{\mathrm{T}}), \tag{6}$$

where $\mathbf{I}_K \in \mathbb{R}^{K \times K}$ is the identity matrix and $\mathbb{1}_K$ is the $K$-dimensional ones vector. $\mathbf{U} \in \mathbb{R}^{d \times K}$ is the patial orthogonal matrix such that $\mathbf{U}^\top \mathbf{U} = \mathbf{I}_K$ and it satisfys $d \geq K$. All vectors in a Simplex ETF have the same pair-wise angle, *i.e.*, $\mathbf{M}_i^{\mathrm{ETF}} \mathbf{M}_j^{\mathrm{ETF}} = -\frac{1}{K-1}, 1 \leq i \neq j \leq K$. The pioneering work [67] shows Simplex ETF as a linear classifier combined with neural networks is robust to class-imbalanced learning in the supervised setting. On the opposite, our motivation is to make self-supervised learning robust to the class-imbalance data, which requires the pursuit in the embedding space intrinsically switching from the sample-level uniformity to the category-level uniformity. The Simplex ETF is a tool to measure the gap between the category-level uniformity and the sample-level uniformity, which is then transformed as the supervision feedback to the training.

### B.2  Alternative Uniform Structure ($K > d$)

For Simplex ETF, there is a hard dimension constraint in Eq. (6), *i.e.*, $K \leq d$. However, if this constraint violates, we do not have such a structure in the hyperspherical space. Alternatively, we can conduct the gradient descent to find an approximation of the maximum separation vertices applied into GH. This refers to minimising the following loss function as demonstrated in [37].

$$\mathcal{L}_{\mathrm{AP}} = \log \sum_{i=1}^{K} \sum_{j=1}^{K} e^{\tilde{\mathbf{M}}_i \cdot \tilde{\mathbf{M}}_j / \tau_{\mathrm{u}}}, \quad \text{s.t.} \quad \sum_{i=1}^{K} \tilde{\mathbf{M}}_i = 0 \text{ and } \forall i \in K \ \|\tilde{\mathbf{M}}_i\| = 1, \tag{7}$$

where the loss term penalizes the pairwise similarity of different vertices [62].

### B.3  Choosing Implementations According to the Dimensional Constraints

As mentioned above, computing the geometric uniform structure $\mathbf{M}$ becomes much harder in the regime of the limited dimension ($K > d$) regarding the hypersphere space [19]. To mitigate this issue, we provide both analytical and approximate solutions for adapting to different application scenarios. Concretely, we choose Simplex ETF (Definition B.1) when $K \leq d$ or the approximated alternatives (Eq. (7)) otherwise. More experimental results can be referred to Appendix F.12.

## C  Theoretical Proofs and Discussions

### C.1  Warmup

We begin by introducing the following lower bound [64] for analyzing the InfoNCE loss.

**Lemma C.1.** *(Lower bound for InfoNCE loss). Assume the labels are one-hot and consistent between positive samples:* $\forall \boldsymbol{x}, \boldsymbol{x}^+ \in p(\boldsymbol{x}, \boldsymbol{x}^+), p(\boldsymbol{y}|\boldsymbol{x}) = p(\boldsymbol{y}|\boldsymbol{x}^+)$. *Let* $\mathcal{L}_{CE}^{\boldsymbol{\mu}}(f) = \mathbb{E}_{p(x,y)} \left[ -\log \frac{\exp(f(\boldsymbol{x})^\top \boldsymbol{\mu}_{\boldsymbol{y}})}{\sum_{i=1}^{K} \exp(f(\boldsymbol{x})^\top \boldsymbol{\mu}_i)} \right]$ *denote the mean CE loss. For* $\forall f \in \mathcal{F}$, *the contrastive learning risk* $\mathcal{L}_{\mathrm{InfoNCE}}(f, \boldsymbol{x}, \boldsymbol{x}^+)$ *can be bounded by the classification risk* $\mathcal{L}_{CE}^{\boldsymbol{\mu}}(f, \boldsymbol{x})$,

$$\mathcal{L}_{\mathrm{InfoNCE}}(f) \geq \mathcal{L}_{\mathrm{CE}}^{\boldsymbol{\mu}}(f) - \sqrt{\mathrm{Var}\,(f(\boldsymbol{x})|\boldsymbol{y})} - \mathcal{O}\left(J^{-\frac{1}{2}}\right) + \log\left(\frac{J}{L}\right) \tag{8}$$

*where* $\sqrt{\mathrm{Var}\,(f(\boldsymbol{x})|\boldsymbol{y})}$ *denotes the conditional intra-class variance* $\mathbb{E}_{p(\boldsymbol{y})} \left[ \mathbb{E}_{p(\boldsymbol{x}|\boldsymbol{y})} \|f(\boldsymbol{x}) - \mathbb{E}_{p(\boldsymbol{x}|\boldsymbol{y})} f(\boldsymbol{x})\|^2 \right]$, $\mathcal{O}\left(J^{-\frac{1}{2}}\right)$ *denotes the Monte Carlo sampling error with* $J$ *samples and* $\log\left(\frac{J}{L}\right)$ *is a constant.*

*Proof.* Let $p(\boldsymbol{x}, \boldsymbol{x}^+, \boldsymbol{y})$ denote the joint distribution $\boldsymbol{x}, \boldsymbol{x}^+$ with the label $\boldsymbol{y}$, $\boldsymbol{y} = 1, \ldots, L$. Denote the negative sample collections as $\{\boldsymbol{x}_i^-\}_{i=1}^J$. According to above assumption on label consistency between positive pairs, we have $\boldsymbol{x}^+$ and $\boldsymbol{x}$ with the same label $\boldsymbol{y}$. Denote $\boldsymbol{\mu}_{\boldsymbol{y}}$ the class means of class $\boldsymbol{y}$ in the embedding space. Then we have the following lower bounds of the InfoNCE loss,

$$\mathcal{L}_{\mathrm{NCE}}(f) = -\mathbb{E}_{p(\boldsymbol{x},\boldsymbol{x}^+)} f(\boldsymbol{x})^\top f(\boldsymbol{x}^+) + \mathbb{E}_{p(\boldsymbol{x})} \mathbb{E}_{p(\boldsymbol{x}_i^-)} \log \sum_{i=1}^{J} \exp(f(\boldsymbol{x})^\top f(\boldsymbol{x}_i^-))$$

$$= -\mathbb{E}_{p(\boldsymbol{x},\boldsymbol{x}^+)} f(\boldsymbol{x})^\top f(\boldsymbol{x}^+) + \mathbb{E}_{p(\boldsymbol{x})} \mathbb{E}_{p(\boldsymbol{x}_i^-)} \log \frac{1}{J} \sum_{i=1}^{J} \exp(f(\boldsymbol{x})^\top f(\boldsymbol{x}_i^-)) + \log J$$

$$\overset{(1)}{\geq} -\mathbb{E}_{p(\boldsymbol{x},\boldsymbol{x}^+)} f(\boldsymbol{x})^\top f(\boldsymbol{x}^+) + \mathbb{E}_{p(\boldsymbol{x})} \log \frac{1}{J} \mathbb{E}_{p(\boldsymbol{x}_i^-)} \sum_{i=1}^{J} \exp(f(\boldsymbol{x})^\top f(\boldsymbol{x}_i^-)) - A(J) + \log J$$

$$= -\mathbb{E}_{p(\boldsymbol{x},\boldsymbol{x}^+)} f(\boldsymbol{x})^\top f(\boldsymbol{x}^+) + \mathbb{E}_{p(\boldsymbol{x})} \log \mathbb{E}_{p(\boldsymbol{x}^-)} \exp(f(\boldsymbol{x})^\top f(\boldsymbol{x}^-)) - A(J) + \log J$$

$$= -\mathbb{E}_{p(\boldsymbol{x},\boldsymbol{x}^+,\boldsymbol{y})} f(\boldsymbol{x})^\top f(\boldsymbol{x}^+) + \mathbb{E}_{p(\boldsymbol{x})} \log \mathbb{E}_{p(\boldsymbol{y}^-)} \mathbb{E}_{p(\boldsymbol{x}^-|\boldsymbol{y}^-)} \exp(f(\boldsymbol{x})^\top f(\boldsymbol{x}^-)) - A(J) + \log J$$

$$\overset{(2)}{\geq} -\mathbb{E}_{p(\boldsymbol{x},\boldsymbol{x}^+,\boldsymbol{y})} f(\boldsymbol{x})^\top f(\boldsymbol{x}^+) + \mathbb{E}_{p(\boldsymbol{x})} \log \mathbb{E}_{p(\boldsymbol{y}^-)} \exp(\mathbb{E}_{p(\boldsymbol{x}^-|\boldsymbol{y}^-)} \left[ f(\boldsymbol{x})^\top f(\boldsymbol{x}^-) \right]) - A(J) + \log J$$

$$= -\mathbb{E}_{p(\boldsymbol{x},\boldsymbol{x}^+,\boldsymbol{y})} f(\boldsymbol{x})^\top (\boldsymbol{\mu}_{\boldsymbol{y}} + f(\boldsymbol{x}^+) - \boldsymbol{\mu}_{\boldsymbol{y}}) + \mathbb{E}_{p(\boldsymbol{x})} \log \mathbb{E}_{p(\boldsymbol{y}^-)} \exp(\mathbb{E}_{p(\boldsymbol{x}^-|\boldsymbol{y}^-)} \left[ f(\boldsymbol{x})^\top f(\boldsymbol{x}^-) \right]) - A(J) + \log J$$

$$= -\mathbb{E}_{p(\boldsymbol{x},\boldsymbol{x}^+,\boldsymbol{y})} [f(\boldsymbol{x})^\top \boldsymbol{\mu}_{\boldsymbol{y}} + f(\boldsymbol{x})^\top (f(\boldsymbol{x}^+) - \boldsymbol{\mu}_{\boldsymbol{y}})] + \mathbb{E}_{p(\boldsymbol{x})} \log \mathbb{E}_{p(\boldsymbol{y}^-)} \exp(f(\boldsymbol{x})^\top \boldsymbol{\mu}_{\boldsymbol{y}^-}) - A(J) + \log J$$

$$\overset{(3)}{\geq} -\mathbb{E}_{p(\boldsymbol{x},\boldsymbol{x}^+,\boldsymbol{y})} \left[ f(\boldsymbol{x})^\top \boldsymbol{\mu}_{\boldsymbol{y}} + \|(f(\boldsymbol{x}^+) - \boldsymbol{\mu}_{\boldsymbol{y}})\| \right] + \mathbb{E}_{p(\boldsymbol{x})} \log \mathbb{E}_{p(\boldsymbol{y}^-)} \exp(f(\boldsymbol{x})^\top \boldsymbol{\mu}_{\boldsymbol{y}^-}) - A(J) + \log J$$

$$\overset{(4)}{\geq} -\mathbb{E}_{p(\boldsymbol{x},\boldsymbol{y})} f(\boldsymbol{x})^\top \boldsymbol{\mu}_{\boldsymbol{y}} - \sqrt{\mathbb{E}_{p(\boldsymbol{x},\boldsymbol{y})} \|f(\boldsymbol{x}) - \boldsymbol{\mu}_{\boldsymbol{y}}\|^2} + \mathbb{E}_{p(\boldsymbol{x})} \log \mathbb{E}_{p(\boldsymbol{y}^-)} \exp(f(\boldsymbol{x})^\top \boldsymbol{\mu}_{\boldsymbol{y}^-}) - A(J) + \log J$$

$$= -\mathbb{E}_{p(\boldsymbol{x},\boldsymbol{y})} f(\boldsymbol{x})^\top \boldsymbol{\mu}_{\boldsymbol{y}} - \sqrt{\mathrm{Var}(f(\boldsymbol{x}) \mid \boldsymbol{y})} + \mathbb{E}_{p(\boldsymbol{x})} \log \frac{1}{L} \sum_{k=1}^{L} \exp(f(\boldsymbol{x})^\top \boldsymbol{\mu}_k) - A(J) + \log J$$

$$= \mathbb{E}_{p(\boldsymbol{x},\boldsymbol{y})} \left[ -f(\boldsymbol{x})^\top \boldsymbol{\mu}_{\boldsymbol{y}} + \log \sum_{k=1}^{L} \exp(f(\boldsymbol{x})^\top \boldsymbol{\mu}_k) \right] - \sqrt{\mathrm{Var}(f(\boldsymbol{x}) \mid \boldsymbol{y})} - A(J) + \log(J/L)$$

$$= \mathcal{L}_{\mathrm{CE}}^\mu(f) - \sqrt{\mathrm{Var}(f(\boldsymbol{x}) \mid \boldsymbol{y})} - A(J) + \log(J/L),$$

where (1) follows Lemma C.2; (2) follows the Jensen's inequality for the convex function $\exp(\cdot)$; (3) follows the hyperspherical distribution $f(\boldsymbol{x}) \in \mathbb{S}^{m-1}$, we have

$$f(\boldsymbol{x})^\top (f(\boldsymbol{x}^+) - \boldsymbol{\mu}_{\boldsymbol{y}}) \leq \left( \frac{f(\boldsymbol{x}^+) - \boldsymbol{\mu}_{\boldsymbol{y}}}{\|f(\boldsymbol{x}^+) - \boldsymbol{\mu}_{\boldsymbol{y}}\|} \right)^\top (f(\boldsymbol{x}^+) - \boldsymbol{\mu}_{\boldsymbol{y}}) = \|f(\boldsymbol{x}^+) - \boldsymbol{\mu}_{\boldsymbol{y}}\|; \tag{9}$$

and (4) follows the Cauchy–Schwarz inequality and the fact that as $p(\boldsymbol{x}, \boldsymbol{x}^+) = p(\boldsymbol{x}^+, \boldsymbol{x})$ holds, $\boldsymbol{x}, \boldsymbol{x}^+$ have the same marginal distribution. $\qquad\square$

In the above proof, the approximation error of the Monte Carlo estimate [64] can be referred to the following lemma.

**Lemma C.2.** *(Upper bound of the approximation error by Monte Carlo estimate) For* LSE $:= \log \mathbb{E}_{p(\boldsymbol{z})} \exp(f(\boldsymbol{x})^\top g(\boldsymbol{z}))$, *we denote its (biased) Monte Carlo estimate with $J$ random samples $\boldsymbol{z}_i \sim p(\boldsymbol{z}), i = 1, \ldots, J$ as $\widehat{\mathrm{LSE}}_J = \log \frac{1}{J} \sum_{i=1}^{J} \exp(f(\boldsymbol{x})^\top g(\boldsymbol{z}_i))$. Then the approximation error $A(J)$ can be upper bounded in expectation as*

$$A(J) := \mathbb{E}_{p(\boldsymbol{x},\boldsymbol{z}_i)} |\widehat{\mathrm{LSE}}(J) - \mathrm{LSE}| \leq \mathcal{O}(J^{-1/2}). \tag{10}$$

*We can see that the approximation error converges to zero in the order of $1/J^{-1/2}$.*

Now we analyze the conditions of Lemma C.1 to strictly achieve its lower bound. In the proof of Lemma C.1, we have four inequality cases and discuss each one as follows:

(1) According to Lemma C.2, we can have the approximation error converges to zero $(A(J) \to 0)$ as the sample population increases to the positive infinity $(J \to +\infty)$. Considering the substantial data amount with regard to the benchmark datasets nowadays, we assume $J$ is large enough and the approximation error can achieve zeros, *i.e.*, $A(J) = 0$.

(2) follows the Jensen's inequality as

$$\mathbb{E}_{p(\boldsymbol{x})} \log \mathbb{E}_{p(\boldsymbol{y}^-)} \mathbb{E}_{p(\boldsymbol{x}^-|\boldsymbol{y}^-)} \exp(f(\boldsymbol{x})^\top f(\boldsymbol{x}^-)) \geq \mathbb{E}_{p(\boldsymbol{x})} \log \mathbb{E}_{p(\boldsymbol{y}^-)} \exp(\mathbb{E}_{p(\boldsymbol{x}^-|\boldsymbol{y}^-)} \left[ f(\boldsymbol{x})^\top f(\boldsymbol{x}^-) \right]). \tag{11}$$

The equality requires the $\exp(\cdot)$ term as a constant:

$$\mathbb{E}_{p(\boldsymbol{x})}\mathbb{E}_{p(\boldsymbol{x}^-)}\exp(f(\boldsymbol{x})^\top f(\boldsymbol{x}^-)) \equiv C_{(2)} \tag{12}$$

(3) The inequality follows

$$f(\boldsymbol{x})^\top(f(\boldsymbol{x}^+) - \boldsymbol{\mu_y}) \leq \left(\frac{f(\boldsymbol{x}^+) - \boldsymbol{\mu_y}}{\|f(\boldsymbol{x}^+) - \boldsymbol{\mu_y}\|}\right)^\top (f(\boldsymbol{x}^+) - \boldsymbol{\mu_y}) = \|f(\boldsymbol{x}^+) - \boldsymbol{\mu_y}\|; \tag{13}$$

where the equality requires $f(\boldsymbol{x})$ has the same direction with $f(\boldsymbol{x}^+) - \boldsymbol{\mu_y}$. Considering the case

$$\mathbb{E}_{p(\boldsymbol{x},\boldsymbol{x}^+,y)}\left[f(\boldsymbol{x}^+) - \boldsymbol{\mu_y}\right] \equiv 0, \tag{14}$$

we should have $\mathbb{E}_{p(\boldsymbol{x},\boldsymbol{x}^+,y)}\left[f(\boldsymbol{x})^\top(f(\boldsymbol{x}^+) - \boldsymbol{\mu_y})\right] \equiv 0$, so the inequality can be simply eliminated from the proof.

(4) Similar in (3), we can simply remove the term $\|(f(\boldsymbol{x}^+) - \boldsymbol{\mu_y})\|$ in $\mathbb{E}_{p(\boldsymbol{x},\boldsymbol{x}^+,y)}\left[f(\boldsymbol{x})^\top\boldsymbol{\mu_y} + \|(f(\boldsymbol{x}^+) - \boldsymbol{\mu_y})\|\right]$ when $\mathbb{E}_{p(\boldsymbol{x},\boldsymbol{x}^+,y)}\left[f(\boldsymbol{x}^+) - \boldsymbol{\mu_y}\right] \equiv 0$.

Note that, Equation (14) requires that all the positive samples approach the class means, *i.e.*, $\forall \boldsymbol{x}^+ \sim p(\boldsymbol{x}^+), f(\boldsymbol{x}^+) = \boldsymbol{\mu_y}$. We then give the following lemma at the state of category-level uniformity.

**Lemma C.3.** *When it satisfies the category-level uniformity (Definition 3.3) defined on the geometric uniform structure* **J** *(Definition 3.1) with dimension $K = L$, assume $A(J) = 0$, for $\forall f \in \mathcal{F}$, the lower bound (Lemma C.1) is achieved as*

$$\mathcal{L}_{\text{InfoNCE}}(f) = \mathcal{L}_{\text{CE}}^\mu(f) + \log\left(\frac{J}{L}\right) \tag{15}$$

*Proof.* According to category-level uniformity (Definition 3.3), we should have

$$\begin{aligned}\mathbb{E}_{p(\boldsymbol{x})}f(\boldsymbol{x}) &\equiv \mathbb{E}_{p(\boldsymbol{x}^+)}f(\boldsymbol{x}^+) \equiv \boldsymbol{\mu}_y, \\ \mathbb{E}_{p(\boldsymbol{x})}\mathbb{E}_{p(\boldsymbol{x}^-)}\exp(f(\boldsymbol{x})^\top f(\boldsymbol{x}^-)) &\equiv C\end{aligned} \tag{16}$$

where the second term is derived from $f(\boldsymbol{x})^\top f(\boldsymbol{x}^-) = \mathbf{M}_i^\top \cdot \mathbf{M}_j = C, i \neq j$ in Definition 3.3. Note that, the category-level uniformity holds on the joint embedding $p(\boldsymbol{x}, \boldsymbol{x}^+)$ of contrastive learning in our setup.

In the proof of Lemma C.1, (1) holds as we assume M is large enough and $A(J) = 0$, (2) holds according to Equation (16), (3)(4) holds as $\mathbb{E}_{p(\boldsymbol{x}^+)}\left[f(\boldsymbol{x}^+) - \boldsymbol{\mu_y}\right] \equiv 0$. As above mentioned, the intra-class variance term $\sqrt{\text{Var}\left(f_\theta(\boldsymbol{x})|\boldsymbol{y}\right)}$ is eliminated. We then have the desired results with Equation (15).

$\square$

## C.2 Proof of Theorem 3.4

*Proof.* On the basis of Lemma C.3, we can derive our overall loss $\mathcal{L}$ as follows,

$$\begin{aligned}\mathcal{L}(f_\theta, \boldsymbol{x}) &= \mathcal{L}_{\text{InfoNCE}}(f_\theta, \boldsymbol{x}, \boldsymbol{x}^+) + \mathcal{L}_{\text{GH}}(f_\theta, \boldsymbol{x}, \hat{\boldsymbol{q}}) \\ &= \mathcal{L}_{\text{CE}}^\mu(f_\theta, \boldsymbol{x}) + \mathcal{L}_{\text{GH}}(f_\theta, \boldsymbol{x}, \hat{\boldsymbol{q}}) + \log\left(\frac{J}{L}\right)\end{aligned} \tag{17}$$

Now we focus on analyzing the minimization of the first and the second term as $\log\left(\frac{J}{L}\right)$ is a constant. Here, we assume the temperature $\gamma_{\text{GH}}$ for generating surrogate labels is small enough, so that we can obtain the discrete geometric labels $\hat{\boldsymbol{q}}$ in one-hot probabilities.

For simplicity, we denote the assigned labels as $t$ for all the data points in class $k$, which are consistent as the samples converge to the class means according to Equation (16). Let $\hat{\mathcal{L}}(f_\theta, \boldsymbol{x}) = \mathcal{L}_{\text{CE}}^\mu(f_\theta, \boldsymbol{x}_k, \boldsymbol{y}) + \mathcal{L}_{\text{GH}}(f_\theta, \boldsymbol{x}_k, t)$, we define the optimization problem regarding class $k$ as:

$$\min \hat{\mathcal{L}}(f_\theta, \boldsymbol{x}_k) = \min \mathcal{L}_{\mathrm{CE}}^{\mu}(f_\theta, \boldsymbol{x}_k) + \mathcal{L}_{\mathrm{GH}}(f_\theta, \boldsymbol{x}_k, t) \tag{18}$$
$$\text{s.t.} \quad \|f_\theta(\boldsymbol{x}_{k,i})\|^2 = 1, \quad \forall i = 1, 2, \ldots, n_k$$

We can then derive

$$
\begin{aligned}
\hat{\mathcal{L}}(f_\theta, \boldsymbol{x}_k) &= \mathcal{L}_{\mathrm{CE}}^{\mu}(f_\theta, \boldsymbol{x}_k) + \mathcal{L}_{\mathrm{GH}}(f_\theta, \boldsymbol{x}_k, t) \\
&= -\frac{1}{n_k} \sum_{i=1}^{n_k} \log \frac{\exp\left(f_\theta(\boldsymbol{x}_{k,i})^\top \cdot \boldsymbol{\mu}_y/\gamma_{\mathrm{CL}}\right)}{\sum_{j=1}^{K} \exp\left(f_\theta(\boldsymbol{x}_{k,i})^\top \cdot \boldsymbol{\mu}_j/\gamma_{\mathrm{CL}}\right)} - \frac{1}{n_k} \sum_{i=1}^{n_k} \log \frac{\exp\left(f_\theta(\boldsymbol{x}_{k,i})^\top \cdot \mathbf{M}_t/\gamma_{\mathrm{GH}}\right)}{\sum_{j=1}^{K} \exp\left(f_\theta(\boldsymbol{x}_{k,i})^\top \cdot \mathbf{M}_j/\gamma_{\mathrm{GH}}\right)} \\
&= -\log \frac{\exp\left(\boldsymbol{\mu}_k^\top \cdot \boldsymbol{\mu}_k/\gamma_{\mathrm{CL}}\right)}{\sum_{j=1}^{K} \exp\left(\boldsymbol{\mu}_k^\top \cdot \boldsymbol{\mu}_j/\gamma_{\mathrm{CL}}\right)} - \log \frac{\exp\left(\boldsymbol{\mu}_k^\top \cdot \mathbf{M}_t/\gamma_{\mathrm{GH}}\right)}{\sum_{j=1}^{K} \exp\left(\boldsymbol{\mu}_k^\top \cdot \mathbf{M}_j/\gamma_{\mathrm{GH}}\right)}
\end{aligned}
\tag{19}
$$

According to Equation (16), the constraints of Equation (18) are equivalent with $\|\boldsymbol{\mu}_k\|^2 = 1$. We can have the Lagrange function as:

$$\tilde{\mathcal{L}} = -\log \frac{\exp\left(\boldsymbol{\mu}_k^\top \cdot \boldsymbol{\mu}_k/\gamma_{\mathrm{CL}}\right)}{\sum_{j=1}^{K} \exp\left(\boldsymbol{\mu}_k^\top \cdot \boldsymbol{\mu}_j/\gamma_{\mathrm{CL}}\right)} - \log \frac{\exp\left(\boldsymbol{\mu}_k^\top \cdot \mathbf{M}_t/\gamma_{\mathrm{GH}}\right)}{\sum_{j=1}^{K} \exp\left(\boldsymbol{\mu}_k^\top \cdot \mathbf{M}_j/\gamma_{\mathrm{GH}}\right)} + \eta_k(\|\boldsymbol{\mu}_k\|^2 - 1) \tag{20}$$

where $\eta_k$ is the Lagrange multiplier.

We consider its gradient with respect to $\boldsymbol{\mu}_k$ as:

$$
\begin{aligned}
\frac{\partial \tilde{\mathcal{L}}}{\partial \boldsymbol{\mu}_k} &= \frac{1}{\gamma_{\mathrm{CL}}} \left[ -(1 - m_k) \cdot \boldsymbol{\mu}_k + \sum_{i \neq k}^{K} m_i \cdot \boldsymbol{\mu}_i \right] + \frac{1}{\gamma_{\mathrm{GH}}} \left[ -(1 - n_k) \cdot \mathbf{M}_t + \sum_{i \neq t}^{K} n_i \cdot \mathbf{M}_i \right] + (\frac{1}{\gamma_{\mathrm{CL}}} + 2\eta_k)\boldsymbol{\mu}_k \\
&= \frac{1}{\gamma_{\mathrm{CL}}} \sum_{i \neq k}^{K} m_i(\boldsymbol{\mu}_i - \boldsymbol{\mu}_k) + \frac{1}{\gamma_{\mathrm{GH}}} \sum_{i \neq t}^{K} n_i(\mathbf{M}_i - \mathbf{M}_t) + (\frac{1}{\gamma_{\mathrm{CL}}} + 2\eta_k)\boldsymbol{\mu}_k
\end{aligned}
\tag{21}
$$

where $m_i = \frac{\exp\left(\boldsymbol{\mu}_k^\top \cdot \boldsymbol{\mu}_i/\gamma_{\mathrm{CL}}\right)}{\sum_{j=1}^{K} \exp\left(\boldsymbol{\mu}_k^\top \cdot \boldsymbol{\mu}_j/\gamma_{\mathrm{CL}}\right)}$, $n_i = \frac{\exp\left(\boldsymbol{\mu}_k^\top \cdot \mathbf{M}_i/\gamma_{\mathrm{GH}}\right)}{\sum_{j=1}^{K} \exp\left(\boldsymbol{\mu}_k^\top \cdot \mathbf{M}_j/\gamma_{\mathrm{GH}}\right)}$.

When it satisfies the category-level uniformity (Definition 3.3) defined on the geometric uniform classifier $\mathbf{M}$ (Definition 3.1), we can obtain $\boldsymbol{\mu}_k = \mathbf{M}_k$.

Multiplying $\mathbf{M}_j$ over the gradients ($j \neq k, j \neq t$):

$$
\begin{aligned}
\frac{\partial \tilde{\mathcal{L}}}{\partial \boldsymbol{\mu}_k} \cdot \mathbf{M}_j &= \frac{1}{\gamma_{\mathrm{CL}}} \sum_{i \neq k} m_i(\boldsymbol{\mu}_i \cdot \mathbf{M}_j - \boldsymbol{\mu}_k \cdot \mathbf{M}_j) + \frac{1}{\gamma_{\mathrm{GH}}} \sum_{i \neq t} n_i(\mathbf{M}_i \cdot \mathbf{M}_j - \mathbf{M}_t \cdot \mathbf{M}_j) + (\frac{1}{\gamma_{\mathrm{CL}}} + 2\eta_k)\boldsymbol{\mu}_k \cdot \mathbf{M}_j \\
&= \frac{1}{\gamma_{\mathrm{CL}}} \sum_{i \neq k} m_i(\mathbf{M}_i \cdot \mathbf{M}_j - \mathbf{M}_k \cdot \mathbf{M}_j) + \frac{1}{\gamma_{\mathrm{GH}}} \sum_{i \neq t} n_i(\mathbf{M}_i \cdot \mathbf{M}_j - \mathbf{M}_t \cdot \mathbf{M}_j) + (\frac{1}{\gamma_{\mathrm{CL}}} + 2\eta_k)\mathbf{M}_k \cdot \mathbf{M}_j \\
&= (m_j + n_j)(1 - C) + (\frac{1}{\gamma_{\mathrm{CL}}} + 2\eta_k)C
\end{aligned}
\tag{22}
$$

where $C$ is defined in Definition 3.1. We can have the probabilities $m_j, n_j$ as

$$m_j = n_j = \frac{1}{1 + (K - 1) \exp\left(C - 1\right)}, \quad j \neq k \tag{23}$$

Let $\eta_k = \frac{C-1}{C+(L-1)C \exp(C-1)} - \frac{1}{2\gamma_{\mathrm{CL}}}$, we can have $\frac{\partial \tilde{\mathcal{L}}}{\partial \boldsymbol{\mu}_k} \cdot \mathbf{M}_j = 0$. With $\mathbf{M}_j \neq 0$, we should have $\frac{\partial \tilde{\mathcal{L}}}{\partial \boldsymbol{\mu}_k} = 0$. Similarly applying to other classes, we can have $\frac{\partial \tilde{\mathcal{L}}}{\partial \boldsymbol{x}} = 0$.

Eventually, we can obtain the minimizer $\hat{\mathcal{L}}^*(f_\theta, x)$ as:

$$\mathcal{L}^*(f_\theta, x) = -\sum_{k=1}^{K} 2\pi_l^y \log\left(\frac{1}{1+(K-1)\exp(C-1)}\right) + \log\left(\frac{J}{L}\right) \qquad (24)$$

$\square$

## C.3   Proof of Lemma 3.2

*Proof.* Assume the samples follow the uniform distribution $n_1 = n_2 = \cdots = n_{L_H} = n_H$, $n_{L_H+1} = n_{L_H+2} = \cdots = n_L = n_T$ in head and tail classes respectively. Assume the imbalance ratio $\frac{n_H}{n_T} \to +\infty$ and the dimenson satisfies $K \geq L$. As proof in [17], we can have

$$\lim \boldsymbol{\mu}_i - \boldsymbol{\mu}_j = \mathbf{0}_L, \ \forall L_H \leq i \leq j \leq L,$$

when the cross-entropy loss achieves the minimizer. Then we can have the lower bound (Lemma C.1) of $\mathcal{L}_{\text{InfoNCE}}$ achieves minimum when the above equation holds, *i.e.*, minority class means collapse to an identical vector.

$\square$

## C.4   Discussions of Lemma 3.2

Intrinsically, Lemma 3.2 is an extreme analysis to characterize the trend under the increasing imbalanced ratios between the majority classes and the minority classes. The staged-wise imbalancing condition is to reach the final compact form about the minority collapse, and more practical long-tailed distribution only reaches the intermediate deduction with much understanding effort, which is even not solved in the current theoretical analysis in supervised long-tailed learning [17]. The $\frac{N_H}{N_t} \to +\infty$ binds with the $\lim$ in the equation is for extreme analysis, but is not for the practical requirement.

## C.5   Applicability of Theorem 3.4

Our theorem and analyses are specific to contrastive learning. In terms of other non-contrastive SSL methods, we empirically show the superiority of our method on long-tailed data distribution in Table 7. Although it might not be straightforward to extend the theory to non-contrastive SSL methods, an explanation about the consistent superiority is that some non-contrastive methods still exhibit similar representation disparity with their contrastive counterpart, and our proposed method can similarly reallocate the geometric distribution to counteract the distorted embedding space. Specially, the recent study [18] theoretically and emprically explore the equivalence between contrastive and non-contrastive criterion, which may shed light on the intrinsic mechanism of how our GH benefits non-contrastive paradigm.

# D   Algorithms

## D.1   Algorithm of Surrogate Label Allocation

We summarize surrogate label allocation in Algorithm 1.

## D.2   Algorithm of Geometric Harmonization

We summarize the complete procedure of our GH method in Algorithm 2.

---

**Algorithm 1** Surrogate Label Allocation.

---

**Input:** geometric cost matrix $\exp(\lambda \log \mathbf{Q})$ with $\mathbf{Q} = [\mathbf{q}_1, \ldots, \mathbf{q}_N]$, marginal distribution constraint $\boldsymbol{\pi}$, Sinkhorn regularization coefficient $\lambda$, Sinkhorn iteration step $E_s$

**Output:** Surrogate label matrix $\hat{\mathbf{Q}}$

1: Set scaling vectors $\boldsymbol{u} \leftarrow \frac{1}{K} \cdot \mathbb{1}_K, \boldsymbol{v} \leftarrow \frac{1}{N} \cdot \mathbb{1}_N$.

2: Set distribution constraints $\boldsymbol{r} \leftarrow \frac{1}{N} \cdot \mathbb{1}_N, \boldsymbol{c} \leftarrow \boldsymbol{\pi}$.

3: **for** iteration $i = 0, 1, \ldots, E_s$ **do**

4: $\quad \boldsymbol{u} \leftarrow \log \boldsymbol{c} - \log\left((\exp(\lambda \log \mathbf{Q})) \cdot \exp(\boldsymbol{v})\right)$.

5: $\quad \boldsymbol{v} \leftarrow \log \boldsymbol{r} - \log\left((\exp(\lambda \log \mathbf{Q}))^\top \cdot \exp(\boldsymbol{u})\right)$.

6: **end for**

7: **return** $\hat{\mathbf{Q}} = N \cdot \mathrm{diag}(\boldsymbol{u}) \exp(\lambda \log \mathbf{Q}) \mathrm{diag}(\boldsymbol{v})$

---

---

**Algorithm 2** Our proposed GH.

---

**Input:** dataset $\mathcal{D}$, number of epochs $E$, number of warm-up epochs $E_w$, geometric uniform classifier $\mathbf{M}$, a self-supervised learning method $\mathcal{A}$

**Output:** pretrained model parameter $\theta_E$

**Initialize:** model parameter $\theta_0$

1: Warm up model $\theta$ for $E_w$ epochs according to $\mathcal{A}$.

2: **for** epoch $e = E_w, E_w + 1, \ldots, E$ **do**

3: $\quad$ Compute the geometric predictions $\mathbf{Q}$ for input samples.

4: $\quad$ Compute the surrogate class prior $\boldsymbol{\pi}$ on training dataset $\mathcal{D}$.

5: $\quad$ **for** mini-batch $k = 1, 2, \ldots, B$ **do**

6: $\quad\quad$ Obtain the surrogate label $\hat{\mathbf{Q}}$ by Algorithm 1.

7: $\quad\quad$ Compute $\mathcal{L}_{\mathrm{CL}}$ according to $\mathcal{A}$ and the proposed $\mathcal{L}_{\mathrm{GH}}$ according to Equation (4).

8: $\quad\quad$ Uptate model $\theta$ by minimizing $\mathcal{L}_{\mathrm{CL}} + \mathcal{L}_{\mathrm{GH}}$.

9: $\quad$ **end for**

10: **end for**

---

# E Supplementary Experimental Setups

## E.1 Dataset Statistics

We conduct experiments on three benchmark datasets for long-tailed learning, including CIFAR-100-LT [5], ImageNet-LT [44] and Places-LT [44]. For small-scale datasets, we adopt the widely-used CIFAR-100-LT with the imbalanced factor of 100, 50 and 10 [5].

In Table 10, we summarize the benchmark datasets used in this paper. Long-tailed versions of CIFAR-100 [34, 16] are constructed following the exponential distribution. For large-scale datasets, ImageNet-LT [44] has 115.8K images with 1000 categories, ranging from 1,280 to 5 in terms of class cardinality and Places-LT [44] contains 62,500 images with 365 categories, with the sample number per category ranging from 4,980 to 5. The large-scale datasets follow Pareto distribution.

As for fine-grained group partitions, we divide each dataset to Many/Medium/Few according to the class cardinality. Concretely, we choose that the largest 34 classes for Many group, the medium 33 classes for Medium group and the smallest 33 classes for Few group on CIFAR-100-LT. On ImageNet-LT and Places-LT, we define Many group with class number over 100, Medium group with 20-100 samples, Few group as under 20 samples [44].

## E.2 Linear probing statistics on the large-scale dataset

The 100-shot evaluation follows the setting in previous works [28, 77]. As shown in Table 11, full-shot evaluation requires 10x - 30x the amount of data compared with the pre-training dataset, which might not be very practical. In contrast, the scale of 100-shot data is consistent with the pre-training dataset. We also present full-shot evaluation in Appendix F.13.

Table 10: Statistics of the benchmark long-tailed datasets. Exp represents exponential distribution.

| Dataset | # Class | Type | Imbalanced Ratio | # Train data | # Test data |
|---|---|---|---|---|---|
| CIFAR-100-LT-R100 | 100 | Exp | 100 | 10847 | 10000 |
| CIFAR-100-LT-R50 | 100 | Exp | 50 | 12608 | 10000 |
| CIFAR-100-LT-R10 | 100 | Exp | 10 | 19573 | 10000 |
| ImageNet-LT | 1000 | Pareto | 256 | 115846 | 50000 |
| Places-LT | 365 | Pareto | 996 | 62500 | 36500 |

Table 11: Statistics of linear probing on the large-scale dataset.

| Dataset | # Class | # Training data | # 100-shot data | # full-shot data | # Test data |
|---|---|---|---|---|---|
| ImageNet-LT | 1000 | 115,846 | 100,000 | 1,261,167 | 50,000 |
| Places-LT | 365 | 62,500 | 36,500 | 1,803,460 | 36,500 |

### E.3 Implementation Details

**Toy Experiments.** We use a 2-Layer ReLU network with 20 hidden units and 2 output units for visualization. For Figure 1, the SimCLR algorithm [9] is adopted in the warm-up stage with proper Gaussian noise as augmentation. After the warm-up stage, we train GH according to Equation (4). We use the orthogonal classifier [(1,1),(-1,1),(-1,-1),(1,-1)] as the geometric uniform structure. For Figure 6, only the SimCLR algorithm is adopted for representation learning.

**More Experimental Setup for Main Results.** (SimCLR, Focal, SDCLR, DnC, BCL) In our experiments, we defaultly set the contrastive learning temperature $\gamma_{\mathrm{CL}}$ as 0.2 and the smoothing coefficient $\beta$ as 0.999 for training stability. For updating the marginal distribution constraint $\pi$, we compute every 20 epochs on CIFAR-100-LT due to the small data size. On ImageNet-LT and Places-LT, we compute $\pi$ every training epoch. Following previous work [28, 77], we adopt a 2-layer MLP as the projector with 128 output dimension. For default data augmentations of contrastive learning, random crop ranging from [0.1, 1], random horizontal flip, color jitter with probability as 0.8 and strength as 0.4 are adopted on CIFAR-100-LT. Random crop ranging from [0.08, 1], random horizontal flip, color jitter with probability as 0.8 and strength as 0.4 and the gaussian blur with probability as 0.5 are adopted on ImageNet-LT and Places-LT.

**Linear Probing Evaluation.** We follow Zhou et al. [77] to conduct Adam optimizer for 500 epochs based on batch size 128, weight decay factor $5 \times 10^{-6}$ and the learning rate decaying from $10^{-2}$ to $10^{-6}$. For few-shot evaluation on ImageNet-LT and Places-LT, we use the same subsampled 100-shot subsets proposed in [77].

### E.4 Focal Loss

Focal loss [40] is discussed and compared in [28, 77] in the context of self-supervised long-tailed learning. Specifically, we use the term inside $\log(\cdot)$ of SimCLR loss as the likelihood to replace the probabilistic term of the supervised Focal loss and obtain the self-supervised Focal loss as:

$$\mathcal{L}_{focal} = -\frac{1}{|\mathcal{D}|} \sum_{\boldsymbol{x} \in \mathcal{D}} (1 - \boldsymbol{p})^{\gamma_{\mathrm{F}}} \log(\boldsymbol{p}), \quad \boldsymbol{p} = \frac{\exp\left(f(\boldsymbol{x})^\top f(\boldsymbol{x}^+)/\gamma_{\mathrm{F}}\right)}{\sum_{x^- \in \mathcal{X}_b^- \cup \{\boldsymbol{x}^+\}} \exp\left(f(\boldsymbol{x})^\top f(\boldsymbol{x}^-)/\gamma_{\mathrm{F}}\right)}$$

where $\gamma_{\mathrm{F}}$ is a temperature factor and $\mathcal{X}_b$ denotes the negative sample set. We defaultly set $\gamma_{\mathrm{F}}$ as 2 in all experiments.

### E.5 Toy Experiments on Various Imbalanced Ratios

In Figure 6, we provide a concrete visualization on a 2-D toy dataset that the sample-level uniformity of the contrastive learning loss leads to the more space invasion of head classes and space collapse of tail classes with increasing the imbalance ratios. According to the results, we can observe that the head classes gradually occupy the embedding space as the imbalanced ratios increase. This further

demonstrates the importance of designing robust self-supervised learning method to counteract the distorted embedding space in the long-tailed context.

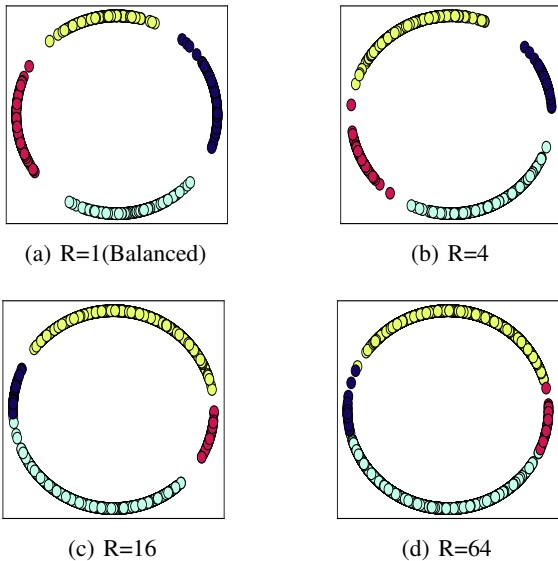

(a) R=1(Balanced)                          (b) R=4

(c) R=16                          (d) R=64

Figure 6: Visualization of the embedding space learnt by vanilla contrastive learning loss on the 2-D imbalanced synthetic dataset with different imbalanced ratios (1,4,16,64). As the ratio increases, head classes gradually occupy the embedding space with the collapse of the tail classes.

# F    Additional Experimental Results and Further Analysis

## F.1    Error Bars for the Main Results

In this part, we present main results with error bars calculated over 5 trials.

Table 12: Linear probing results (average accuracy, %) over 5 trials on CIFAR-LT with different imbalanced ratios (100,50,10), ImageNet-LT and Places-LT.

|         | CIFAR-LT-R100 | CIFAR-LT-R50 | CIFAR-LT-R10 | ImageNet-LT | Places-LT |
|---------|---------------|--------------|--------------|-------------|-----------|
| SimCLR  | 50.72±0.26    | 52.24±0.31   | 55.67±0.44   | 36.65±0.16  | 33.61±0.12 |
| +GH     | 53.96±0.23    | 55.42±0.22   | 57.36±0.39   | 38.28±0.13  | 34.33±0.10 |
| Focal   | 51.04±0.27    | 52.22±0.38   | 56.23±0.45   | 37.49±0.11  | 33.65±0.14 |
| +GH     | 53.92±0.19    | 55.06±0.28   | 58.05±0.28   | 38.92±0.14  | 34.42±0.17 |
| SDCLR   | 52.87±0.22    | 53.87±0.21   | 55.44±0.25   | 36.25±0.18  | 33.99±0.14 |
| +GH     | 54.81±0.26    | 55.34±0.28   | 56.97±0.34   | 38.53±0.14  | 34.70±0.10 |
| DnC     | 52.52±0.32    | 53.21±0.35   | 57.59±0.36   | 37.23±0.21  | 33.90±0.18 |
| +GH     | 54.88±0.23    | 56.33±0.31   | 58.94±0.25   | 38.67±0.19  | 34.52±0.23 |
| BCL     | 56.45±0.40    | 57.18±0.26   | 59.12±0.28   | 38.33±0.10  | 34.76±0.15 |
| +GH     | 57.65±0.33    | 59.00±0.33   | 60.34±0.29   | 39.95±0.15  | 35.32±0.17 |

## F.2    Convergence of the Surrogate Label Allocation

In Table 13, we provide the experiments to verify the convergence of the Sinkhorn-Knopp algorithm, which adopts the criterion as the stopping reference.

We define the criterion $e = sum(|u./u' - 1|)$ as the relative changes of one scaling vectors $u$, where $u'$ represents the vector in the latest iteration. Then, the algorithm converges as the criterion $e \to 0$.

Table 13: The value of $e$ during the convergence of surrogate label allocation on CIFAR-LT-R100.

| Iter | 0 | 10 | 20 | 30 | 50 | 70 | 100 | 150 |
|------|------|------|------|-------|--------|--------|---------------------|---------------------|
| $e$ | 67.89 | 4.28 | 0.53 | 0.076 | 0.0054 | 0.0005 | $2.08 \times 10^{-5}$ | $3.58 \times 10^{-7}$ |

As shown in Table 13, we can see that the criterion diminishes rapidly. Let $e < 10^{-6}$ represent the indicator of the convergence, we further obtain the averaging convergence iterations as $141 \pm 45$ (statistics under 1000 runs). In practice, we set the default Sinkhorn iterations as 300 to guarantee the convergence, as detailed in Section 4.1.

### F.3 Empirical Comparison with More Baselines

In Table 14, we conduct a range of experiments to compare PMSN[35] and TS [2] with our proposed GH on CIFAR-LT with different imbalanced ratios.

Table 14: Linear probing accuracy of more SSL-LT baselines on CIFAR-100-LT with different imbalanced ratios.

| | Method | Many | Med | Few | Avg |
|---|---|---|---|---|---|
| CIFAR-R100 | SimCLR | 54.97 | 49.39 | 47.67 | 50.72 |
| | SimCLR+TS | 55.53 | 50.33 | 50.06 | 52.01 |
| | PMSN | 55.62 | 52.12 | 49.85 | 52.56 |
| | SimCLR**+GH** | 57.38 | 52.27 | **52.12** | 53.96 |
| | SimCLR+TS**+GH** | **57.44** | **52.76** | 51.79 | **54.03** |
| CIFAR-R50 | SimCLR | 56.00 | 50.48 | 50.12 | 52.24 |
| | SimCLR+TS | 56.44 | 52.58 | 51.91 | 53.67 |
| | PMSN | 56.76 | 52.52 | 53.09 | 54.15 |
| | SimCLR**+GH** | **58.88** | 53.00 | 54.27 | 55.42 |
| | SimCLR+TS**+GH** | 58.47 | **54.61** | **54.70** | **55.95** |
| CIFAR-R10 | SimCLR | 57.85 | 55.06 | 54.03 | 55.67 |
| | SimCLR+TS | 58.26 | 56.24 | 54.97 | 56.51 |
| | PMSN | 56.91 | 54.61 | 55.67 | 55.74 |
| | SimCLR**+GH** | 59.26 | 56.91 | 55.85 | 57.36 |
| | SimCLR+TS**+GH** | **59.44** | **57.15** | **56.48** | **57.71** |

From the results, we can see that the proposed method consistently outperforms PMSN[35] and TS [2] across different imbalanced ratios on CIFAR-LT. Besides, we can observe that combining GH and TS [2] consistently improves the performance of contrastive learning on CIFAR-LT.

### F.4 Empirical Comparison with K-Means Algorithm

K-means algorithm [22] tends to generate clusters with relatively uniform sizes, which will affect the cluster performance under the class-imbalanced scenarios [38]. To gain more insights, we conduct empirical comparisons using K-means as the clustering algorithm and evaluate the NMI score with ground-truth labels and the linear probing accuracy on CIFAR-LT-R100.

Table 15: Linear probing accuracy and NMI score on CIFAR-100-LT-R100.

| Method | Accuracy | NMI score |
|---|---|---|
| SimCLR | 50.72 | 0.28 |
| +K-means | 51.44 | 0.35 |
| +GH | 53.96 | 0.50 |

From the results, we can see that K-means generates undesired assignments with lower NMI score and achieves unsatisfying performance compared with our GH. This observation is consistent with previous studies [38].

## F.5 Compatibility on the Class-Balanced Data

In Table 16, we present the results on the balanced dataset CIFAR-100 across different methods.

Table 16: Linear probing on class-balanced CIFAR-100. We report Accuracy(%) for comparison.

| Method | SimCLR | +GH | Focal | +GH | SDCLR | +GH | DnC | +GH | BCL | +GH |
|---|---|---|---|---|---|---|---|---|---|---|
| Accuracy | 66.75 | 66.41 | 66.42 | 66.79 | 65.46 | 66.17 | 67.78 | 67.57 | 69.16 | 69.33 |

From the results, we can see that GH shows comparable performance with the baseline methods when the data distribution is balanced. According to the neural collapse theory [49], well-trained neural networks can inherently produce the category-level uniformity on class-balanced data. As expected, our GH will degenerate to the vanilla SSL baselines as the geometric labels can easily be aligned with the latent ground-truth labels. The empirical findings are also consistent with recent explorations [31] in supervised learning context. Besides, the minor decrease in performance could potentially be attributed to some random factors during training or the negligible effect of GH loss as it might not reach an absolute zero value.

## F.6 Computational Cost

In Table 17, we present the mini-batch training time of different baseline methods on CIFAR-100-LT, ImageNet-LT and Places-LT.

Table 17: The time cost (seconds) of mini-batch training on CIFAR-100-LT, ImageNet-LT and Places-LT.

| Dataset | SimCLR | +GH | Focal | +GH | SDCLR | +GH | DnC | +GH | BCL | +GH |
|---|---|---|---|---|---|---|---|---|---|---|
| CIFAR-LT | 0.38 | 0.41 | 0.37 | 0.40 | 0.42 | 0.47 | 0.39 | 0.41 | 0.38 | 0.41 |
| ImageNet-LT | 0.76 | 0.79 | 0.75 | 0.77 | 0.94 | 1.01 | 0.76 | 0.79 | 0.76 | 0.78 |
| Places-LT | 0.72 | 0.75 | 0.76 | 0.78 | 1.00 | 1.05 | 0.72 | 0.76 | 0.72 | 0.75 |

In our runs, the proposed GH only incurs a minor computational overhead on CIFAR-100-LT, ImageNet-LT and Places-LT, respectively, which is relatively lightweight compared to the total computational cost of the contrastive baselines. This indicates the great potential of GH to collaborate with more SSL methods to acquire the robustness on data imbalance in a low-cost manner.

## F.7 Ablations on Hyper-parameters

In this part, we present ablation studies *w.r.t.* temperature $\gamma_{\text{GH}}$, regularization coefficient $\lambda$ and Sinkhorn iteration $E_s$ on CIFAR-LT.

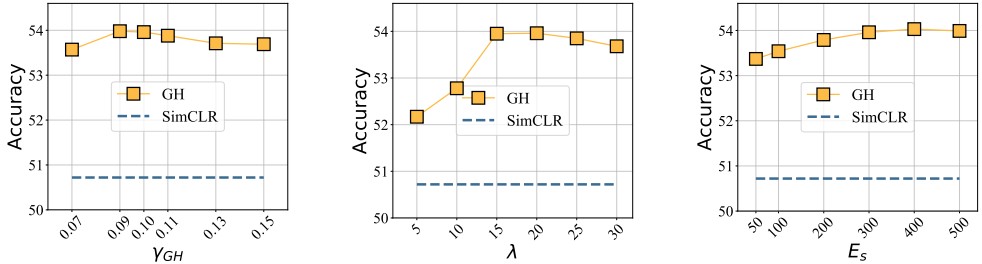

Figure 7: Ablations of temperature $\gamma_{\text{GH}}$, coefficient $\lambda$ and Sinkhorn iteration $E_s$ on CIFAR-LT.

## F.8 Ablations on the Training Epoch

In Table 18, we present the comparison between SimCLR and SimCLR+GH on CIFAR-LT-R100 under different training epochs.

Table 18: Linear probing results on CIFAR-LT-R100 with different training epochs.

| epoch | 200 | 500 | 1000 | 1500 | 2000 |
|---|---|---|---|---|---|
| SimCLR | 49.53 | 50.32 | 50.72 | 50.84 | 50.27 |
| +GH | 50.89 | 54.00 | 53.96 | 53.95 | 53.91 |

According to the table, we can see that both methods appropriately reach the saturated performance when the training epochs are larger than 500. To guarantee the converged performance, we thus set the default training epochs as 1000.

## F.9 Ablations on the Batch Size

To explore the effect of the training batch size, we conduct the experiments with different batch size on CIFAR-LT-R100 as follows.

Table 19: Linear probing results on CIFAR-LT-R100 w.r.t the methods with different batch size.

| Batch size | 128 | 256 | 512 | 768 | 1024 |
|---|---|---|---|---|---|
| SimCLR | 50.14 | 51.08 | 50.72 | 50.25 | 50.07 |
| +GH | 52.72 | 53.43 | 53.96 | 53.95 | 53.18 |

Table 20: Linear probing results on ImageNet-LT w.r.t the methods with different batch size.

| Batch size | 256 | 384 | 512 | 768 |
|---|---|---|---|---|
| SimCLR | 36.65 | 36.97 | 37.85 | 38.04 |
| +GH | 38.28 | 39.22 | 41.06 | 41.34 |

From the results, we can see that our GH consistently outperforms the baseline SimCLR. It is worth noting that our method still provides siginificant improvements when the batch size is small (e.g. 2.6% with batch size as 128 on CIFAR-LT), which reflects the robustness of the proposed GH in terms of small batch sizes. Besides, we observe that the performance drops when reducing the batch size for both baseline method and our GH on CIFAR-LT and ImageNet-LT, as shown in Table 19. This can potentially be attributed to the higher probability of encountering situations where certain classes are missing under smaller batch size. Intuitively, it might easily generate biased estimation when there is no support for a certain class in the mini-batch. Then, the cluster quality might be affected by the probability of encountering missing class, which potentially correlates the important factor, *i.e.*, batch size.

## F.10 Ablations on Geometric Uniform Structure

In Table 21, we conduct experiments with the geometric uniform structure as the projector on top of the baseline contrastive learning methods. As can be seen, if geometric uniform structure alone is used to balance the representation learning, the improvement is minor and sometimes degrades. This is because the direct estimation from the geometric uniform structure is noisy during training when the representation is not ideally distributed.

Table 21: Ablations of the geometric uniform structure on CIFAR-100-LT with different imbalanced ratios (100, 50, 10).

| Method | CIFAR-LT-R100 | CIFAR-LT-R50 | CIFAR-LT-R10 |
|---|---|---|---|
| SimCLR | 50.72 | 52.24 | 55.67 |
| +GUS | 51.10 | 51.99 | 55.56 |
| +GH | 53.96 | 55.42 | 57.36 |

### F.11 Ablations on the Momentum Hyper-parameter

In our proposed GH, the hyper-parameter $\beta$ controls the smoothing degree on the historical statistics regarding the dynamically estimated surrogate label distribution $\pi$. We conduct empirical comparison with different $\beta$ to validate the stability of our method, as depicted in Figure 8. From the results, we can see that our GH can achieve consistent performance at the most cases. To guarantee the performance, we thus set the default hyper-parameter $\beta$ as 0.999.

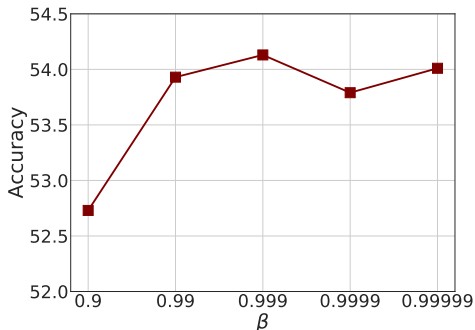

Figure 8: Linear probing *w.r.t* hyper-parameter $\beta$ on CIFAR-LT-R100.

### F.12 Implementations of Geometric Uniform Structure

In Figure 9, we empirically compare the results of analytical geometric uniform structure (Simplex ETF) with those of proxy variants. We thus conduct the sensitivity analysis w.r.t a smaller span of K, ranging from 30 to 220. From the results, we observe that the comparable performance is achieved in both geometric structures. This indicates that our method is effective to two forms of the geometric structure, relaxing the hard dimensional constraints in the analytical solution. It is also worth noting that our method's efficacy remains unaffected by the dimension of the geometric uniform structure when appropriately choosing the dimension, highlighting its ease of application.

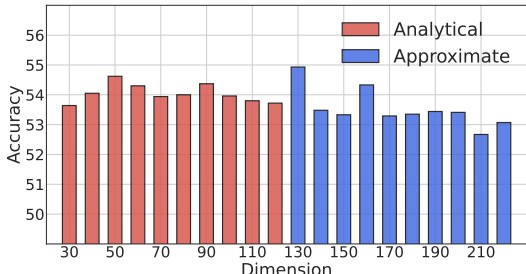

Figure 9: Linear probing performance *w.r.t.* the dimension $K$ of the geometric uniform structure $\mathbf{M}$ on CIFAR-LT-R100. Analytical or approximate solution are applied according to the dimensional constraints. More details can be referred to Appendix B.

### F.13 Full-shot Evaluation on Large-scale Dataset

Here we provide 100-shot evaluation and full-shot evaluation on ImageNet-LT, as shown in Table 22. We observe that the performance improvements and representation balancedness (Std) are consistent with both evaluations, indicating the rationality of the 100-shot evaluation.

### F.14 Comprehensive Evaluation on More Real-world Scenarios

To further validate the generalization of the proposed method, we conduct more comprehensive comparisons on various datasets with distinct characteristics and tasks, and conduct more experiments as follows:

Table 22: Full-shot linear evaluation and 100-shot evaluation on ImageNet-LT.

| Evaluation | Method | Many | Medium | Few | Std | Avg |
|---|---|---|---|---|---|---|
| 100-shot | SimCLR | 41.69 | 33.96 | 31.82 | 5.19 | 36.65 |
|  | +GH | 41.53 | 36.35 | 35.84 | 3.15 | 38.28 |
| Full-shot | SimCLR | 42.86 | 35.17 | 33.13 | 5.13 | 37.86 |
|  | +GH | 44.11 | 38.59 | 37.87 | 3.41 | 40.62 |

- Marine-tree dataset [3]: This dataset is a large-scale dataset for marine organism classification. It contains more than 160K images divided into 60 classes with the number of images per class ranging from 14 to 16761.
- IMDB-WIKI-DIR dataset [69]: IMDB-WIKI-DIR (age) dataset is subsampled from IMDB-WIKI dataset [51] to construct the deep imbalanced regression benchmark. It contains 202.5K images with the number of images per bin varied between 1 and 7149.
- CUB-200 [61] and Aircrafts [45] dataset. Caltech-UCSD Birds 200 (CUB-200) and Aircrafts dataset are two fine-grained datasets, which contains 11K images with 200 classes and 10K images with 102 classes, respectively.

Table 23: Linear probing results (average accuracy, %) on Marine-tree dataset.

| Marine | Many | Medium | Few | Avg |
|---|---|---|---|---|
| SimCLR | 36.05 | 47.01 | 48.80 | 43.95 |
| +GH | 35.70 | 47.14 | 51.62 | 44.82 |

Table 24: Vanilla finetuning results under the metric of mean average error (MAE [69], **lower is better**) on IMDB-WIKI-DIR dataset.

| IMDB-WIKI-DIR (MAE) | Many | Medium | Few | Avg |
|---|---|---|---|---|
| SimCLR | 8.10 | 18.31 | 29.99 | 9.14 |
| +GH | 7.77 | 17.18 | 29.29 | 8.75 |

Table 25: Vanilla finetuning results under the metric of Geometric Mean (GM [69], **lower is better**) on IMDB-WIKI-DIR dataset.

| IMDB-WIKI-DIR (GM) | Many | Medium | Few | Avg |
|---|---|---|---|---|
| SimCLR | 4.87 | 15.01 | 26.61 | 5.43 |
| +GH | 4.64 | 13.49 | 24.54 | 5.14 |

Table 26: Downstream linear probing results (Top1/Top5 accuracy, %) on CUB-200 dataset.

| CUB-200 | SimCLR | +GH | Focal | +GH | SDCLR | +GH | DnC | +GH | BCL | +GH |
|---|---|---|---|---|---|---|---|---|---|---|
| TOP1 | 28.97 | 29.89 | 30.13 | 30.68 | 28.98 | 29.63 | 29.64 | 30.46 | 28.46 | 28.97 |
| TOP5 | 57.28 | 57.92 | 58.01 | 58.78 | 57.34 | 57.95 | 57.63 | 58.55 | 56.92 | 57.66 |

Table 27: Downstream linear probing results (Top1/Top5 accuracy, %) on Aircrafts dataset.

| Aircrafts | SimCLR | +GH | Focal | +GH | SDCLR | +GH | DnC | +GH | BCL | +GH |
|---|---|---|---|---|---|---|---|---|---|---|
| TOP1 | 29.82 | 30.63 | 31.02 | 31.74 | 30.99 | 31.85 | 31.18 | 32.05 | 32.79 | 35.88 |
| TOP5 | 56.14 | 57.95 | 57.82 | 58.99 | 58.09 | 59.13 | 58.11 | 59.42 | 60.79 | 63.34 |

On large-scale dataset (Marine-tree dataset and IMDB-WIKI-DIR dataset), we adopt the training schedule similar to ImageNet-LT and Places-LT, except the training epochs reduced from 500 to

about 200 epochs. Besides, we crop the images with the low resolution (112x112) to speed up the training. We conduct linear probing on Marine-tree, CUB-200 and Aircrafts dataset. The former is pretrained with Marine-tree dataset, while the latter is pretrained with ImageNet-LT. As for IMDB-WIKI-DIR dataset, we pretrain the network for initializing the weights of the downstream supervised imbalanced regression task. Specially, the geometric mean (GM) is defined for better prediction fairness [69]. Both the evaluation metrics (MAE, GM) are the smaller the better.

From the results in Tables 23 to 27, we can see that our proposed GH consistently outperforms the baseline methods for all the metrics (linear probing accuracy, finetuning accuracy, MAE and GM) on various datasets/settings. This indicates the potential of GH for adapting to a wide range of real-world data scenarios to counteract the negative impact of the long-tailed distribution.

### F.15 More Results on Joint Optimization with Warm-up Strategy

We can potentially adopt warm-up strategy to initialize the weights $\theta$ against the degenerate solutions in the joint optimization. In this subsection, we conduct more comprehenvive experiments on CIFAR-LT-100 with different warm-up epochs to further verify the superiority of the proposed bi-level optimization.

Table 28: Linear probing results of joint optimization on CIFAR-LT-R100 with different warm-up epochs.

| Epoch | 0 | 10 | 50 | 100 | 200 | 300 | 400 |
|---|---|---|---|---|---|---|---|
| Accuracy | 50.18 | 51.14 | 50.77 | 50.97 | 50.44 | 50.21 | 50.57 |

From the results in Table 28, we can see that the warm-up strategy has the potential to improve the linear probing performance by 1% over the vanilla joint training. However, it seems that this strategy is sensitive to the proper epochs for warming-up, and the overall performance is not better than the bi-level optimization.

## G  Broader Impacts

Learning long-tailed data without annotations is a vital element in the deployment of robust deep learning systems in the real-world applications [25, 76, 72, 73]. The attribution is that real-world natural resources inevitably exhibit the long-tailed distribution [50]. The importance of self-supervised long-tailed learning is further emphasized when extended to a range of safety-critical scenarios [8, 26, 27], including medical intelligence [65, 66, 74], autonomous driving [55–57] and criminal surveillance, where the data imbalance may lead to the distorted representation. In this paper, we study a general and practical research problem in representation learning parity for self-supervised long-tailed learning, considering the intrinsic limitation of conventional contrastive learning that can not adequately address the over-expansion of the majorities and the passive-collapse of the minorities in the embedding space. Our method regularizes long-tailed learning from a geometric perspective and motivates more benign representation, which helps improve the downstream generalization and representation balancedness. Besides, our method has the potential to be applied in fairness research scenarios [42] where both majority and minority classes (or attributes) are present. Given the guidance of label information, we can explicitly constrain a consistent embedding space for each subgroup, thereby promoting category-level uniformity.

Nevertheless, it is important to acknowledge that our method may have negative impact, such as employment disruption, as our study endeavors to reduce annotation costs by enabling robust self-supervised learning on hard-to-collect tail data resources. Specially, if self-supervised learning can extract the tail distribution with sufficient accuracy, the necessity for the human manipulation on the quality of data distribution will diminish.

## H  Limitations

Roughly, our design is built upon the intrinsic clustering patterns that can inclusively represent the information for the downstream tasks. Although we demonstrate the appealing performance in the

current benchmark, it cannot be always guaranteed in all scenarios. Once such a condition is not satisfied, namely, clustering only captures the task-irrelevant patterns but ignores the task-relevant details, the improvement might be limited or even negative. A potential way to overcome this drawback is using a small auxiliary labeling set to calibrate the clustering dynamic aligned with the downstream tasks, namely, a semi-supervised paradigm. The methods to encourage learning the stable features in the area of causal inference can also be borrowed to this problem to alleviate this dilemma.