# OpenReview forum: "Combating Representation Learning Disparity with Geometric Harmonization"
_NeurIPS.cc/2023/Conference — NeurIPS 2023 spotlight_

### Official Review · Reviewer_R4di · 2023-06-14

**Soundness:** 2 fair
**Presentation:** 2 fair
**Contribution:** 3 good
**Rating:** 6
**Confidence:** 2

**Summary:**

The authors tackle the problem of contrastive learning in the context of imbalanced datasets where some classes have many more samples than other classes. Since no label information can be used, importance sampling is not an option and without any interventions, contrastive learning tends to overrepresent majority classes in the latent space, thereby hurting the readout accuracy on the minority classes. The authors propose a Geometric Harmonization technique in which they propose to estimate the class labels with clustering and then recalibrate the learned embeddings such that majority classes occupy the same feature space as minority classes. The authors show experimental results on CIFAR100-LT, ImageNet-LT and Places-LT.

**Strengths:**

- I appreciate the complexity analysis of the proposed method in Section 3.4.
- The authors compare to several other benchmarks and show superior results with their method.
- I think in general the method makes sense and it is intuitive that it should work in the proposed case.


**Weaknesses:**

- There are many typos and grammatical errors in the text which makes the paper hard to understand. Some mathematical definitions are wrong and inconsistent which makes me doubt the claims of the theoretical guarantees since there are even errors in the Lemmas / definitions:

     - Line 94: “Let N denote the sample number and R = max_i n_i/min_i n_i denote the imbalanced ratio, where n_i denotes the number of samples in class i.” I am confused, the max and min are calculated per class i, which means that max_i n_i = min_i n_i = n_i, therefore R would be 1 for each class. Since I do not understand how R is defined, I cannot understand / judge the results in Table 2.
     - Lemma 3.2 is inconsistent with itself. If n_L_H=n_H and n_L_H=n_T, then n_T = n_H and n_H / n_T -> infty is impossible because n_H / n_T = 1 by definition.
    - Eq. 5: Using M both as the geometric structure and the size of the collection of negative samples is confusing and should not be done.

- Considering the experimental results, I am not convinced that the authors have looked into the most important baselines. CIFAR100-LT, ImageNet-LT and Places-LT are well-established benchmarks and the baseline results that the authors aim to improve seem to be very low to me. I will elaborate on the different datasets in the following:
     - For ImageNet-LT (https://paperswithcode.com/sota/long-tail-learning-on-imagenet-lt), the currently best number for a ResNet50 architecture is an accuracy of 70.1 (https://arxiv.org/pdf/2111.13579v4.pdf) with extra training data and 67.2 (https://arxiv.org/pdf/2111.14745v1.pdf) without extra training data. The best result the authors report in this paper is about 38% which is far below what is currently state of the art according to the benchmark list.
     - For Places-LT, https://paperswithcode.com/sota/long-tail-learning-on-places-lt, the best numbers for a ResNet50 are above 45% while the authors here report numbers below 35%.

-> I believe that in order for this method to be relevant to the community, the authors need to show results on the superior models with higher baseline accuracy. In the current state, it is not clear whether their results would generalize to the better models.

- The motivation for the studied problem does not become clear to me from the introduction. The authors write: “However, the real-world natural sources usually exhibit the long-tailed distribution [31], and directly learning representation on them might lead to the distortion issue of the embedding space, namely, the majority dominates the feature regime [45] and the minority collapses [28].” Using the modal verb “might” here indicates a possibility but no further evidence is presented. Citing fairness research as an application is not enough in my opinion. From reading the introduction, I am not convinced that the problem the authors want to study actually exists in SSL. I would advise the authors to provide concrete examples where using SSL actually harms performance on minority classes. Currently, the main question of study posed in line 39 (“Why the conventional contrastive learning underperforms in self-supervised long-tailed context?”) does not seem well supported.

- The first part of the first contribution is misleading and I believe wrong: “To our best knowledge, we are the first to investigate the drawback of the contrastive learning loss in self-supervised long-tailed context.” This drawback is investigated in several other papers which the authors discuss in their related work section, e.g. SDCLR specifically tackles this problem with a different method.

- Figure 1: The spheres in the middle and right Figure look like ovals which is confusing. These should be spheres/ circles.

- Abstract, line 5: „The attribution is that the vanilla SSL methods that pursue the sample-level uniformity easily leads to representation learning disparity, where head classes with the huge sample number dominate the feature regime but tail classes with the small sample number passively collapse.” -> This sentence is hard to parse and understand.




- Table 1: What does IR stand for?

- Line 119: “and π ∈ RK+ refers to the the marginal distribution constraint.” -> "The" twice and what do you mean under “marginal distribution constraint”?

- The used datasets and models must be cited.


- Line 162: “This phenomenon indicates all representations of the minority will collapse completely to one point without considering the category discrepancy, which corresponds to our observation regarding the passive collapse of tailed samples in Figure 1.” This statement is not accurate because the representations of the minority classes do not collapse to a single point in Figure 1. Lemma 3.2. covers an extreme case where there are infinitely more samples in the head classes compared to the tail classes, so Figure 1 does not represent this scenario and thus, using “corresponds” here is not accurate.



- It is confusing that Table 7 comes before Tables 5+6, please fix.

- Does Table 5 show the computational cost per epoch?

- Line 218: “For hyper-parameters of GH, we provide a default setup across all the experiments: set the geometric dimension K as 100 and the temperature γGH as 0.1. In the surrogate label allocation, we set the regularization coefficient λ as 20 and Sinkhorn iterations Es as 300. Please refer to Appendix E.3 for more experimental details.” There are many hyperparameters that need to be set for this method, and it is not clear whether they were chosen on the test set, or how they were selected. It is also not clear how sensitive the algorithm is to these hyperparameters.

- Line 223: “For comprehensive performance comparison, we present the linear probing performance and the standard deviation among three disjoint groups, i.e., [many, medium, few] partitions [25].” Please explain what the partitions into many/medium/few mean as it is not possible to understand the results otherwise.

- Line 290: “To justify the intuition in Section 3.2” which intuition? Please be more specific.

- I find the results in Fig. 3b unintuitive. The NMI scores show that GH is better aligned with ground truth labels compared to all other methods, but then why is the readout accuracy of GH only 1-2 percent points better compared to the other methods? Please discuss this as I am not sure how to interpret this result and whether it is meaningful to compare NMI scores between GH and the other methods.


**Questions:**

- How sensitive is GH to the choice of the hyperparameters?
- Can you please comment on how the presented results on Places-LT and ImageNet-LT compare to the state-of-the-art results on https://paperswithcode.com/sota/long-tail-learning-on-places-lt and https://paperswithcode.com/sota/long-tail-learning-on-imagenet-lt?

**Limitations:**

The authors could think a bit more about the impact of their work on applications related to long-tailed data distributions where minority and majority classes are present. How would applications situated in fairness research impacted by their work?

---

> ### Author Rebuttal · Authors · 2023-08-09
>
> While the reviewer raised so many questions, we sincerely appreciate the reviewer's time and effort on our submission. Probably due to the difference in small research directions, the reviewer has some misunderstanding in the setting, background and evaluation of this topic. We do hope that the following point-to-point responses can address the concerns raised by the reviewer. Any further comments are welcomed.
>
> > **Q1:** Typos, grammatical errors and notation usage.
>
> **A1:** Sorry for the misunderstanding due to some typos, the definition $R$ and the notation usage. We will carefully address each of then in the revision to ease the understanding. Specially, we would like to explain that $R = N_{max}/N_{min}$ denotes the imbalance ratio, where $N_{max}$,$N_{min}$ represents the sample number in the largest\/smallest class, and will revise the mentioned notation as $n_{L_H+1}=n_{L_H+2}=\dots=n_{L}=n_T$, and use $J$ denote the number of negative samples.
>
> > **Q2:** Supervised long-tailed benchmark.
>
> **A2:** Thanks for the question. We would like to kindly clarify the differences between self-supervised long-tailed (SSL-LT) learning and the recommended supervised long-tailed learning (S-LT). Firstly, SSL-LT does not rely on any label information in the training stage, while S-LT leverages the fully supervised data. Furthermore, SSL-LT aims to address the issue of representation disparity, pursuing more balanced embedding space. In contrast, S-LT focuses on the classification disparity, which involves rectifying both the representation and the last-layer classifier. This distinction encourages different evaluation and corresponding performance of two paradigms. Specially, in SSL-LT, linear probing is employed to evluate the quality of representation, while classification is adopted to evaluate the accuracy of the predictions in S-LT. Consequently, direct performance comparison between SSL-LT and S-LT may not be feasible or appropriate.
>
> > **Q3:** Motivation.
>
> **A3:** Thanks for the questions. We would like to kindly clarify on the evidence regarding the limitation of contrastive learning on long-tailed data as follows:
>
> - Linear probing evidence. Table 2 shows that many-shot classes outperforms few-shot classes by an average of 5.67\%/9.87\% on CIFAR-LT/ImageNet-LT with SimCLR [11]. This indicates that contrastive loss is not immune to class imbalance, leading to performance disparities. Our empirical findings are consistent with several recent SSL-LT explorations [14,16,20,21].
> - Cross-dataset transfer. We conduct experiments on large-scale long-tailed CC3M and evaluated on various downstream tasks (See **Table R3-R4**). The results reveals that representation disparity actually exists in real-world data and GH promotes better transferability.
> - Uniformity metric. **Table R5** reveals that contrastive learning struggles to achieve the class-uniform partition, while our method effectively mitigates this issue.
> - Qualitive analysis. **Figure 1,R1** show that contrastive learning leads to representation disparity, where head classes dominate the embedding space and tail classes exhibit passive collapse.
>
> We will enhance the explanation about the difference of two sub-areas to avoid misunderstanding.
>
> > **Q4:** Clarification on first contribution.
>
> **A4:** Thanks for the question. We would like to reclarify our taxonomy as described in **Lines 28-38** that all previous explorations target to improve self-supervised long-tailed learning, but from different aspects like reweight/optimization techniques, architecture design and data augmentation. However, **none of these methods focus on the intrinsic limitation of the contrastive learning**. Concretely, we provide more detailed aspects in **Table G1** in the general response for clarity.
>
>
>
> > **Q5:** Minor issues: (1) Oval-like embedding space in Figure 1, (2) long sentence in line 5, (3) twice "the" in line 162, (4) cite dataset and model, (5) “correspond” in line 162 (6) Table 7 location. (7) intuitions in line 290.
>
> **A5:** Thanks for your detailed suggestions. We will carefully follow the reviewer's advice to revise each of them in the revision.
>
> > **Q6:** Questions: (1) IR in Table 1, (2) marginal distribution constraint in line 119, (3) computational cost in Table 5, (4) many/medium/few partitions
>
> **A6:** Thanks for your detailed questions. Below are explanations to each questions:
>
> - IR indicates the imbalanced ratio $R$.
> - It means the distribution prior to determine the marginal projections of matrix $\hat{Q}$ onto its rows and columns.
> - Table 5 shows the time cost per mini-batch. (See caption in Table 5)
> - Please refer to line 692-696 in Section E.1 in appendix for partition details.
>
> > **Q7:** Sensitivity to the hyper-parameters.
>
> **A7:** Thanks for the question. We would like to kindly clarify that many important hyperparameters have been compared or discussed (See Figure 3(a) and Table 11-13, Figure 5). Besides, we follow reviewer's advice to conduct more experiments on temperature $\gamma_{GH}$, coefficient $\lambda$ and Sinkhorn iteration $E_s$ on CIFAR-LT. The results in **Figure R3** shows that GH consistently achieves satisfying performance with different hyperparameter.
>
> > **Q8:** NMI and linear probing.
>
> **A8:** Thanks for the question. Below are the detailed reasons:
>
> - Linear probing removes the projector and GUS, while NMI uses their predictions.
> - NMI evaluates on train set, while linear probing evaluates on test set.
> - NMI is the normalization of the mutual information, while linear probing reports accuracy.
>
> These differences make the scale for the relative improvement in NMI cannot be compared with that for accuracy.
>
> > **Q9:** Applications in fairness research.
>
> **A9:** Thanks for the constructive suggestions. We will add more discussions about how GH can benefit the applications situated in fairness research, especially from the perspective of representation parity, and cite the related works.

---

> > ### Comment · Reviewer_R4di · 2023-08-11
> > **Response to the rebuttal**
> >
> > Dear authors,
> >
> > thank you for your replies to my concerns. It is indeed that I am not familiar with the literature on self-supervised long-tailed (SSL-LT) learning, as well as with the field in general, as is reflected by my low confidence. Given that other much confident reviewers write that the experimental results are strong, I will lift this concern. I will increase my rating to a '6', but will not increase my confidence.
> >
> > I encourage the authors to improve the clarity of the paper, rectify the definitions etc. My low rating was also due to me not understanding several important equations because they were wrong / very ambiguous.
> >
> > Best,
> > Reviewer R4di

---

> > > ### Author Response · Authors · 2023-08-11
> > > **Thank you for the response**
> > >
> > > Dear Reviewer R4di,
> > >
> > > We sincerely appreciate you taking the time to review our responses and contributing to improve this paper. We will follow reviewer's advice to thoroughly proofread the paper again to enhance its clarity and facilitate understanding. We will also carefully incorporate all the addressed points in the updated version. Thank you once again for your dedicated and valuable contribution in reviewing our paper!
> > >
> > > Best,
> > >
> > > Authors of Paper 11308

---

### Official Review · Reviewer_G4d9 · 2023-06-30

**Soundness:** 2 fair
**Presentation:** 3 good
**Contribution:** 2 fair
**Rating:** 6
**Confidence:** 4

**Summary:**

The paper investigates the SSL in a long-tailed distribution setting, where traditional contrastive learning may not work well.
The authors attribute that issue to the "sample-level" learning of SSL and propose Geometric Harmonization (GH) to regularize the "category-level" during training. GH can be a plug-and-play loss to the existing SSL frameworks without modification.
Empirical results of linear probing accuracy show the effectiveness of the proposed method in the long-tailed data over the baseline SimCLR.

**Strengths:**

+ Contribute to SSL capability in real-world scenarios with imbalanced classes which may be useful for unsupervised representation learning.
+ Some theoretical analyses are provided to support the problem and method.
+ The idea is reasonable
+ Writing is clear enough

**Weaknesses:**

+ SSL has been developed with various methods to facilitate the representation learning for unlabelled data, this paper provided the experiments for one method, i.e. SimCLR, which is not a very strong baseline and not a state-of-the-art CL framework as claimed.
From the presented results, SimCLR still works not too bad compared to SimCLR+GH in all settings. I would expect to see the effectiveness of GH in more strong SSL baselines including contrastive learning frameworks such as BYOL, MoCo-v3, etc, and maybe a more recent branch of SSL, namely mask autoencoder (MAE). To make the proposed method more complete, those baselines should be considered.
+ The "passive collapse" of tail classes is mentioned in the paper but seems that except for some quantitative results showing GH improves baseline to some extent, there are no results (either visualization or quantitative metric) to show such collapse happens and that GH can mitigate or solve that problem.

**Questions:**

+ For transfer learning to downstream tasks, which dataset is pre-trained for SSL?
+ From the concept/hypothesis in Fig. 2 bottom, when applying GH, the samples of each class are clustered well even though it is the imbalanced case. Is there visualization (t-SNE for example) for the real data that has been conducted in this paper to show that works as the figure of concept in Fig.2?

**Limitations:**

They have discussed the limitations.

---

> ### Author Rebuttal · Authors · 2023-08-09
>
> We gratefully thank you for your time and efforts devoted to reviewing this paper and your constructive suggestions. Here are our detailed replies to your questions.
>
> > **Q1:** Empirical studies on BYOL, MoCo-v3 and maybe a more recent branch of SSL, namely mask autoencoder (MAE).
>
> **A1:** Thanks for the constructive suggestions. We would like to kindly clarify that many representative self-supervised learning methods have been empirically compared in the submission, such as SimCLR [11], SeLa [17], SwAV[18], MoCo-v2[22], SimSiam[23] and Barlow Twins[24] (See Table 1,4). The comprehensive comparison of different SSL baselines includes (1) contrastive learning: SimCLR and MoCo-v2, (2) unsupervised clustering: SeLa and SwAV and (3) some other non-contrastive methods: SimSiam, Barlow Twins. Besides, we do appreciate the reviewer’s advice about conducting comparisons on more SSL baselines, and conduct more experiments based on BYOL [25] and MoCo-v3 [26] on CIFAR-LT with different imbalanced ratios.
>
> From the results in **Table R1** in the complementary PDF file, we can see that GH can consistently improve BYOL/MoCo-v3 across different imbalanced ratios on CIFAR-LT on imbalanced datasets. Due to the time constraints, we will provide more comprehensive experiments/discussions to strengthen their comparisons with MAE [27] and on ImageNet in the revision.
>
> > **Q2&Q4:** Visualization or quantitative metric demonstrating the passive collapse of tail classes and t-SNE visualization to support the concept/hypothesis presented in Fig. 2 on read data.
>
> **A2&A4:** We appreciate the reviewer's constructive suggestions, and conduct both quantitative and qualitive analyses to provide further intuitions into the proposed method.
>
> **Quantitative analysis:**
>
> We conduct more thorough experiments with several metrics [28,29] on CIFAR-LT as follows:
>
> - Inter-Class Uniformity: $U=\frac{1}{L(L-1)}\sum_{i=1}^L\sum_{j=1, j\neq i}^L||\mu_i-\mu_j||_2$, where we have $L$ classes and $\mu$ denotes the class means. It evaluates average distances between different class centers.
> - Neighborhood Uniformity: $U_k=\frac{1}{Lk}\sum_{i=1}^L\min_{j_1,\cdots, j_k}(\sum_{m=1}^k||\mu_i-\mu_{j_m}||_2)$, where $j_1, \cdots, j_k \neq i$ represent different classes. It measures how close one class is to its neighbors.
>
> In **Table R5** in the complementary PDF file, we compare these metrics on CIFAR-LT with different imbalanced ratios, and have the following observations: our GH exhibits significant improvements in both inter-class uniformity and neighborhood uniformity when compared with the baseline SimCLR. This indicates that vanilla contrastive learning struggles to achieve the uniform partitioning of the embedding space, while our proposed method effectively mitigates this issue.
>
> **Qualitive analysis:**
>
> We conduct t-SNE visualization of SimCLR and SimCLR with GH on CIFAR-LT-R100. For simplity, we randomly selected four head classes and four tail classes to generate the t-SNE plots. Based on the results in **Figure R1** in the complementary PDF file, the observations are as follows: (1) SimCLR: head classes exhibit a large presence in the embedding space and heavily squeeze the tail classes, (2) GH: head classes reduce their occupancy, allowing the tail classes to have more space.
>
> This indicates that passive collapse of tail classes still persists in real-world data and our method can effectively mitigate the negative effect. Moreover, our method demonstrates the ability to promote clusters with higher quality in the presence of class-imbalanced data, surpassing the baseline SimCLR. We will include these discussion and empirical quantitive/quanlitive comparisons in the revision.
>
> > **Q3:** For transfer learning to downstream tasks, which dataset is pre-trained for SSL?
>
> Thanks for the detailed question. In section 4.3, the SSL methods are pretrained on the same datasets as the downstream datasets, i.e., ImageNet-LT to ImageNet-LT and Places-LT to Places-LT. In section F.9 in the appendix, the SSL methods are pretrained on ImageNet and linear probing evaluated on the CUB200 [4] and Aircraft [5].
>
> Besides, we also conduct more comprehensive experiments on various cross-dataset transferring tasks to evaluate representation transferability. Specifically, we pretrain our method on large-scale long-tailed dataset CC3M [1]. Subsequently, we apply our approach to various downstream classification datasets, including well-curated datasets such as ImageNet [2] and Places [3], fine-grained datasets such as CUB200 [4], AirCraft [5], Stanford Cars [6], Stanford Dogs [7] and NABirds [8]. Furthermore, we also conduct empirical comparisons on object detection and segmentation task using COCO2017. From the results in **Table R3-R4** in the complementary PDF file, we can see that our proposed GH consistently outperforms the baseline across various tasks and datasets. This indicates the superiority of GH to improve contrastive learning on imbalanced datasets, yielding better model generalization to a range of real-world scenarios. We appreciate the reviwer's question and will additional clarifications and empirical studies on transfer learning in the revision.

---

> > ### Comment · Reviewer_G4d9 · 2023-08-11
> >
> > I have read the rebuttal and other reviews. Basically it addressed adequately my concerns, and I would love to raise the score. I recommend the authors add these addressed points to their revision.

---

> > > ### Author Response · Authors · 2023-08-11
> > > **Thank you for the response**
> > >
> > > Dear Reviewer G4d9,
> > >
> > > We sincerely appreciate you taking the time to review our responses and contributing to improve this paper. We will carefully follow reviewer's advice to incorporate all the addressed points in the updated version.
> > >
> > > Best,
> > >
> > > Authors of Paper 11308

---

### Official Review · Reviewer_pT3P · 2023-07-06

**Soundness:** 3 good
**Presentation:** 3 good
**Contribution:** 3 good
**Rating:** 7
**Confidence:** 3

**Summary:**

- The paper proposes to tackle the task of self-supervised learning on datasets sampled from long-tailed distributions.
- Typical SSL approaches tend to perform a lot worse for the minority classes, thus hurting performance.
- The proposed approaches termed Geometric Harmonization encourages category-level uniformity leading to better performance on various downstream tasks.
- The approach is shown to be complementary to various existing SSL approaches.

**Strengths:**

- The paper tackles an important problem in SSL - that of training on data sampled from long-tailed distributions which is common in the real world.
- The proposal of geometric harmonization is interesting. The idea is simple and easy to implement. While the approach is motivated by recent clustering-based works in SSL, the proposed approach seems novel.
- The paper is well written for the most part (some concerns are pointed out in other sections)
- Authors perform extensive analyses and ablations to understand the working of the approach.
- Experiments on various LT datasets show the benefit of using the approach.

**Weaknesses:**

Following are some of my concerns with the paper:
- Clarity: Given that geometric harmonization is the key contribution of this work, I think the authors should give the readers more idea about the intuitions for the section on "surrogate label allocation". While the Figure 2 looks good, the caption is not very helpful to drive home the intuition. I see a similar issue with Definition 3.1 : meaning of K only becomes clear later in the paper.
- Effect of batch-size : The authors have provided an analysis on the effect of batch size. I think such analysis will be especially interesting for datasets which have lot more labels : like ImageNet.
- Discussion/comparison missing on some related works: [a], [b]


[a] The hidden uniform cluster prior in self-supervised learning
[b] Temperature Schedules for Self-Supervised Contrastive Methods on Long-Tail Data

**Questions:**

- Minor suggestions :
    - L42 : reword "as proof in [38].."
    - "sample number" : I suggest using "number of samples" or a similar alternative instead to avoid any confusion.
 - Some more discussion and analysis on the observation in L298-300 would be really interesting. For this experiment, what happens if there is no support for a certain class in that mini-batch ?


**Limitations:**

The authors have adequately addressed the limitations and societal impact of their work.

---

> ### Author Rebuttal · Authors · 2023-08-09
>
> We gratefully thank you for your time and efforts devoted to reviewing this paper and your constructive suggestions. Here are our detailed replies to your questions.
>
> > **Q1:** Clarity: Given that geometric harmonization is the key contribution of this work, I think the authors should give the readers more idea about the intuitions for the section on "surrogate label allocation". While the Figure 2 looks good, the caption is not very helpful to drive home the intuition. I see a similar issue with Definition 3.1 : meaning of K only becomes clear later in the paper.
>
> **A1:** We appreciate the reviewer's constructive suggestions. We will add additional clarifications regarding the "surrogate label allocation" section especially about how and why it works in self-supervised long-tailed context. Futhermore, we will thoroughly proofread the paper again to enhance its clarity and facilitate understanding including the caption of Figure 2 and $K$ in Definition 3.1.
>
>
> > **Q2:** Effect of batch-size : The authors have provided an analysis on the effect of batch size. I think such analysis will be especially interesting for datasets which have lot more labels : like ImageNet.
>
> **A2:** Thanks for the advice. We summarize the experiments under different batch sizes on ImageNet as follows.
>
>
>
> | Batchsize |  256  |  384  |  512  | 768 |
> |:---------:|:-----:|:-----:|:-----:|:---:|
> |  SimCLR [11]   | 36.65 |  36.97     |  37.85     | 38.04    |
> |    +GH    | 38.28 | 39.22 | 41.06 | 41.34    |
>
>
> From the results, we can see that under different batch sizes, our GH consistently outperforms the baseline method and a larger batch size relatively promotes better performance. We will include detailed experiments and discussions regarding the training batch size on large-scale datasets in the revision. Specially, we refer the reviewer to A5, which explains why a smaller batch size may hurt the performance under the long-tailed data from many categories.
>
>
> > **Q3:** Discussion/comparison missing on some related works: [a], [b].
>
> > [a] The hidden uniform cluster prior in self-supervised learning
> > [b] Temperature Schedules for Self-Supervised Contrastive Methods on Long-Tail Data
>
> **A3:** We very appreciate the reviewer to recommend the concurrent works PMSN [19] and TS [20], and will add these works into related works with the proper discussion.
>
>
>
> Regarding the comparison, we conduct a range of experiments on CIFAR-LT with different imabalanced ratios to compare PMSN/TS with GH as shown in **Table R2** in the complementary PDF file.
>
>
> From the results, we can see that the proposed method consistently outperforms PMSN/TS across different imbalanced ratios on CIFAR-LT. Besides, we can observe that combining GH and TS consistently improves the performance of contrastive learning on CIFAR-LT. Due to the time constraints, we will provide more comprehensive experiments/discussions to strengthen their comparisons in the revision.
>
>
> > **Q4:** Minor suggestions
>
> > (1) L42 : reword "as proof in [38].."
> > (2) "sample number" : I suggest using "number of samples" or a similar alternative instead to avoid any confusion.
>
> **A4:** Thanks for your detailed suggestions. We will carefully revise each of them and proofread the submission.
>
> > **Q5:** Some more discussion and analysis on the observation in L298-300 would be really interesting. For this experiment, what happens if there is no support for a certain class in that mini-batch?
>
> **A5:** This is a very insightful question and thank the reviewer for this point. Intuitively, it might easily generate biased estimation when there is no support for a certain class in the mini-batch.Then, the cluster quality might be affected by the probability of encountering missing class, which potentially correlates one important factor, i.e., batch size. Empirically, as demonstrated in Table 13 in Appendix F.4, we obeserve that the performance drops when reducing the batch size by a factor of 4 on CIFAR-LT. This can potentially be attributed to the higher probability of encountering situations where certain classes are missing under smaller batch size. We appreciate the valuable question and will include more detailed experiments and discussion in the revision.

---

> > ### Comment · Reviewer_pT3P · 2023-08-14
> > **Thanks for the rebuttal**
> >
> > Thanks for the very detailed rebuttal. The explanations and new experiments on CC3M are very helpful. I have no additional concerns at this moment.

---

> > > ### Author Response · Authors · 2023-08-14
> > > **Thank you for the response**
> > >
> > > Dear Reviewer pT3P,
> > >
> > > We sincerely appreciate you taking the time to review our responses and contributing to improve this paper. We will incorporate the mentioned points in the updated version. Thank you once again for your dedicated and valuable contribution!
> > >
> > > Best,
> > >
> > > Authors of Paper 11308

---

### Official Review · Reviewer_TBJn · 2023-07-08

**Soundness:** 3 good
**Presentation:** 3 good
**Contribution:** 3 good
**Rating:** 7
**Confidence:** 4

**Summary:**

This paper concentrates on the long-tailed distribution problem in SSL representation learning. To overcome the challenge in vanilla SSL methods, where head classes with the huge sample number dominate the feature regime, the authors propose the Geometric Harmonization (GH) method to encourage category-level uniformity. Specially, GH measures the population statistics of the embedding space, and then infer an fine-grained instance-wise calibration to constrain the space expansion of head classes while avoid the collapse of tail classes. Extensive results show the effectiveness of GH with high tolerance to the long-tailed distribution problem.

**Strengths:**

-1- This paper investigates the drawback of the contrastive learning loss in SSL long-tailed context, and shows that the resulted sample-level uniformity is an intrinsic limitation to the representation parity, as shown in Figure 1. The motivation of pursuing category-level uniformity is clear.

-2- This paper is well-written and easy to follow. The proposed GH method that incorporates the geometric clues from the embedding space to calibrate the training loss, is interesting and solid.

-3-  The proposed GH loss is versatile and can be easily plugged into existing SSL methods. The extensive experiments and ablation stidies demonstrate its effectiveness in learning robust representation.

**Weaknesses:**

-1- About the surrogate label allocation. What is the advantage to choose the discriminative clustering [1], what if we choose other clustering method such as K-means. How does the quality of the geometric label affect the final training results? When verifing the assumption that the constructed surrogate geometric labels are mutually correlated with the oracle labels, can the authors provide more intuitive results?

-2- In E.q. 4, do you need to design how to balance these two losses?

-3- Table 6 shows the results on class-balanced data. Can you explain why in some cases, incooperateing with GH will lead to the performance drop?

**Questions:**

Please refer to *Weaknesses.

**Limitations:**

Yes.

---

> ### Author Rebuttal · Authors · 2023-08-09
>
> We gratefully thank you for your time and efforts devoted to reviewing this paper and your constructive suggestions. Here are our detailed replies to your questions.
>
> > **Q1:** About the surrogate label allocation. What is the advantage to choose the discriminative clustering [1], what if we choose other clustering method such as K-means. How does the quality of the geometric label affect the final training results? When verifing the assumption that the constructed surrogate geometric labels are mutually correlated with the oracle labels, can the authors provide more intuitive results?
>
> **A1:** We appreciate the reviewer's insightful questions. Below are our replies to each subquestion.
>
> **Advantage:** Our discriminative clustering approach offers the following advantages: (1) it enables a flexible class prior that can adapt to various distributions, (2) it refines assignments in an efficient manner, (3) it is theoretically grounded to achieve the desired category-level uniformity along with the geometric uniform structure, effectively addressing the challenges posed by the long-tailed effect. Note that, we would like to reclarify that straightforward applying the discriminative clustering SeLA [17] or the variant SwAV [18] cannot work, whose difference from ours is discussed in Lines 134-152, and empirically verified in Table 1 of the original submission.
>
> **Drawback of K-means:** K-means algorithm tends to generate clusters with relatively uniform sizes, which will affect the cluster performance under the class-imbalanced scenarios [19]. To gain more insights, we conduct empirical comparisons using K-means as the clustering algorithm and evaluate the NMI score with ground-truth labels and the linear probing accuracy on CIFAR-LT-R100.
>
> | Method | Accuracy | NMI score |
> |:--|:--|:--|
> | SimCLR [11] | 50.72 | 0.28 |
> | +K-means | 51.44 | 0.35 |
> | +GH | 53.96 | 0.50 |
>
> From the results, we can see that K-means generates undesired assignments with lower NMI score and achieves unsatisfying performance compared with our GH. This observation is consistent with previous studies [19].
>
> **More intuitive results:** To enhance the understanding of the proposed surrogate label allocation, we conduct both quantitative and qualitive analyses.
>
> (1) Quantitative analyses:
>
> We conduct more thorough experiments with several metrics [28,29] on CIFAR-LT, including:
>
> - Inter-Class Uniformity: $U=\frac{1}{L(L-1)}\sum_{i=1}^L\sum_{j=1, j\neq i}^L||\mu_i-\mu_j||_2$, where we have $L$ classes and $\mu$ denotes the class means. It evaluates average distances between different class centers.
> - Neighborhood Uniformity: $U_k=\frac{1}{Lk}\sum_{i=1}^L\min_{j_1,\cdots, j_k}(\sum_{m=1}^k||\mu_i-\mu_{j_m}||_2)$, where $j_1, \cdots, j_k \neq i$ represent different classes. It measures how close one class is to its neighbors.
>
> Based on the results in **Table R5** in the complementary PDF file, the observations are as follows: our method exhibits significant improvements in both inter-class uniformity and neighborhood uniformity when compared with the baseline SimCLR [11]. This indicates that vanilla contrastive learning struggles to achieve the uniform partitioning of the embedding space, while our proposed method effectively mitigates this issue.
>
> (2) Qualitive analyses:
>
> We conduct t-SNE visualization of SimCLR and SimCLR with GH on CIFAR-LT. For simplity, we randomly selected four head classes and four tail classes to generate the t-SNE plots. Based on the results in **Figure R1** in the complementary PDF file, the observations are as follows: (1) SimCLR: head classes exhibit a large presence in the embedding space and heavily squeeze the tail classes, (2) GH: head classes reduce their occupancy, allowing the tail classes to have more space. This further indicates that the constructed surrogate labels can serve as the high-quality supervision, effectively guiding the harmonization towards the geometric uniform structure.
>
>
> > **Q2:** In E.q. 4, do you need to design how to balance these two losses?
>
> **A2:** In the submission, we have not introduced a hyperparameter to balance two losses, instead, we defaultly set a weight 1.0 (termed as $w_{GH}$) on GH loss in Eq.4 across all the experimental results. To address the question regarding balancing contrastive loss and our GH loss, we conduct experiments with different weight $w_{GH}$ on CIFAR-LT-R100 (please refer to **Figure R2** in the complementary PDF file).
>
> From the results, we can see that our method generally achieves comparable performance across different configurations of weight $w_{GH}$. We appreciate the reviewer's question and will add the discussion and empirical comparisons in the revision.
>
> > **Q3:** Table 6 shows the results on class-balanced data. Can you explain why in some cases, incooperateing with GH will lead to the performance drop?
>
> **A3:** Thanks for your detailed question. We would like to kindly clarify that our GH is generally comparable with baseline methods, which aligns with our expectations. Specifically, our GH is designed to offer advantages on imbalanced datasets while avoiding unreasonable degradation to contrastive learning on balanced dataset. In Table 6, the proposed GH shows the minor improvements over Focal [12], SDCLR [14] and BCL [16], and performs worse than SimCLR  and DnC [15] for less than 0.3\%. The minor decrease in performance could potentially be attributed to some random factors during training (like weight initialization and data augmentation) or the negligible effect of GH loss as it might not reach an absolute zero value.

---

> > ### Comment · Reviewer_TBJn · 2023-08-11
> > **Response to Author Rebuttal**
> >
> > Thanks for the response that addressed my concerns, and I will keep my positive score.

---

> > > ### Author Response · Authors · 2023-08-11
> > > **Thank you for the response**
> > >
> > > Dear Reviewer TBJn,
> > >
> > > We sincerely appreciate you taking the time to review our responses and contributing to improve this paper. We will carefully follow reviewer's advice to incorporate all the addressed points in the updated version.
> > >
> > > Best,
> > >
> > > Authors of Paper 11308

---

### Official Review · Reviewer_M9kH · 2023-07-10

**Soundness:** 3 good
**Presentation:** 4 excellent
**Contribution:** 3 good
**Rating:** 7
**Confidence:** 4

**Summary:**

This paper studied self-supervised representation learning with implicit class-imbalance data, in contrast to most prior works that assume class-balance. To do this, this paper proposed to augment the existing contrastive learning method with a novel geometric harmonization (GH). Intuitively, GH pulls samples toward the closest prototype in the embedding space, which differs from previous cluster-/prototype-based methods in the learnable prototypes and marginal prior. Through extensive experiments on several standard long-tail classification benchmarks, the authors demonstrated the efficacy of the proposed method with different contrastive learning methods as start points.

**Strengths:**

1. The paper is generally well-written and easy to follow.

2. The paper is well-motivated. Class imbalance occurs in most real-world scenarios, and existing contrastive learning methods fail to take this into account. This paper provides a good exploration of this direction.

3. The proposed geometric harmonization regularization is interesting. Though prototype-based constraints are already studied in contrastive learning [1, 5, 12], GH carefully considers the class imbalance by its design and is thus robust to such imbalance.

4. The experiment results are strong compared with the baselines on imbalanced data. Moreover, the proposed method even performs on par with the baseline with balanced data.

**Weaknesses:**

1. The geometric harmonization relies heavily on the assumption that samples from the same class intrinsically lie in neighboring regions in the embedding space to estimate the assignment and prior. While this condition can be satisfied on curated datasets like CIFAR and ImageNet, it might be a different case for the uncurated datasets, e.g., YMCC or a subset of it. Some analysis on this end would further strengthen this paper.

2. Lack of transfer learning experiments. In all the experiments, the models were pretrained and then finetuned on the same datasets. It is unclear whether the learned representations are general to transfer to other data distribution and different tasks. Possible options are adding cross-dataset experiments (e.g., YFCC-to-ImageNet-LT, ImageNet-Lt-to-CIFAR-LT) and transfer learning on MS-COCO object detection as in [5].

**Questions:**

See the weakness section.

**Limitations:**

The major limitation is that geometric harmonization relies heavily on the assumption that samples from the same class intrinsically lie in neighboring regions in the embedding space, which may not hold for real-world data.

---

> ### Author Rebuttal · Authors · 2023-08-09
>
>
> We gratefully thank you for your time and efforts devoted to reviewing this paper and your constructive suggestions. Here are our detailed replies to your questions.
>
> > **Q1:** The geometric harmonization relies heavily on the assumption that samples from the same class intrinsically lie in neighboring regions in the embedding space to estimate the assignment and prior. While this condition can be satisfied on curated datasets like CIFAR and ImageNet, it might be a different case for the uncurated datasets, e.g., YMCC or a subset of it. Some analysis on this end would further strengthen this paper.
>
>
> **A1:** This is a very insightful question and thank the reviewer for this point. We agree with the reviewer for the potential presence of scattered samples from the same classes across different regions (rather than neighboring regions) in the embedding space, particularly in the context of large-scale long-tailed data distributions. In this case, our method will leverage the inherent pattern or semantic cluster information of the data in the fine-grained region to relatively mitigate the disparity. Such potential provides the model a chance to learn more general-purpose representations correlated to the clusters in various levels of granularities, going beyond the observed labels, thus promoting the generality and transferability of the pretrained representations. We do appreciate the reviewer's advice and conduct experiments on **large-scale long-tailed and uncurated dataset CC3M** [1] to address the possible concern, as detailed in **A2**, and we will include the discussion into the submission.
>
>
> > **Q2:** Lack of transfer learning experiments. In all the experiments, the models were pretrained and then finetuned on the same datasets. It is unclear whether the learned representations are general to transfer to other data distribution and different tasks. Possible options are adding cross-dataset experiments (e.g., YFCC-to-ImageNet-LT, ImageNet-Lt-to-CIFAR-LT) and transfer learning on MS-COCO object detection as in [5].
>
> **A2:** We appreciate the reviewer’s constructive advice and conduct more comprehensive experiments on various cross-dataset transferring tasks with distinct characteristics as follows:
>
>
> - Pretraining datasets
>     - CC3M [1]. CC3M is a **large-scale, long-tailed** dataset with more than 3 million images. We pretrain SimCLR [11] and SimCLR with GH on CC3M for comparison.
> - Transferring to downstream classification
>     - Curated datasets: ImageNet [2] and Places [3]. We randomly subsample a balanced subset of ImageNet and Places for downstream finetuning, with the number of images per class set as 100.
>     - Fine-grained datasets: CUB200 [4], AirCraft [5], Stanford Cars [6], Stanford Dogs [7], NABirds [8]. These datasets requires semantic features to distinguish each category on the fine-grained granularity.
>     - We report the average accuracy of our CC3M-pretrained weights finetuning on classification on these datasets based on ResNet50 backbone.
> - Transferring to downstream object detection
>     - COCO2017 [9]. We report the bounding box AP of our CC3M-pretrained weights finetuning on object detection on COCO2017 using Faster-RCNN based on ResNet50-FPN backbone.
> - Transferring to downstream segmentation
>     - COCO2017 [9]. We report the mask AP of our CC3M-pretrained weights finetuning on segmentation on COCO2017 using Mask-RCNN based on ResNet50-FPN backbone.
>
>
>
>
> From the results in **Table R3-R4** in the complementary PDF file, we can see that our proposed GH consistently outperforms the baseline methods across various tasks and datasets. This indicates the superiority of GH to improve contrastive learning on imbalanced datasets, yielding better model generalization to a range of real-world scenarios.
>
> Besides, for more complete comparisons under different baselines, we will update the results in the submission once we finish all experiments.

---

> > ### Comment · Reviewer_M9kH · 2023-08-18
> > **Response to the authors**
> >
> > Thanks for the great effort in the rebuttal. It resolves my concerns about the premise and the transferability of this method. I have no additional concerns now and would like to raise the score.

---

> > > ### Author Response · Authors · 2023-08-19
> > > **Thank you for the response**
> > >
> > > Dear Reviewer M9kH,
> > >
> > > We sincerely appreciate you taking the time to review our responses and contributing to improve this paper. We will incorporate all the addressed points in the updated version. Thank you once again for your dedicated and valuable contribution in reviewing our paper!
> > >
> > > Best,
> > >
> > > Authors of Paper 11308

---

### Official Review · Reviewer_krnL · 2023-07-18

**Soundness:** 3 good
**Presentation:** 3 good
**Contribution:** 3 good
**Rating:** 6
**Confidence:** 4

**Summary:**

This paper addresses the class-imbalance problem under the SSL setting. It shows that the Vanilla SSL methods pursuing the sample-level uniformity easily lead to representation learning disparity, where head classes with the huge sample number dominate the feature regime but tail classes with the small sample number passively collapse. It proposes Geometric Harmonization (GH) to encourage the category-level uniformity in representation learning. The GH loss can be easily integrated into existing SSL methods, and GH loss improve the performance over baselines on the imbalance class problem under the SSL setting, based on the experimental results.

**Strengths:**

1.This paper define the category-level uniformity in the embedding space that SSL learns.  The motivation to using category-level uniformity for imbalance class problem is clear.

2.The proposed Geometric Harmonization (GH) loss improve the performance over baselines on the imbalance class problem under the SSL setting, based on the experimental results.

3.The presentation of this paper is generally clear and easy to follow.

**Weaknesses:**

1. In the experiments of Section 4.3, it is better to conduct experiments on large-scale ImageNet pertaining using the proposed methods, and then transfer to other datasets (This is the so called “representation transferability”), rather than only: SSL pre-training and finetuning on the same datasets.

2. I personally believe the theories and analyses both are specific to contrastive learning, rather than on other SSL methods (e.g., the non-contrastive SSL methods, BOYL, SiamSiam). Even though this paper conducts an experiment on other SSL methods in Section 4.4, it is not clear how the theory/analyses work for these SSL methods? Do the theoretic analyses still work for other SSL methods? This paper should at least mention it.

3. At last, I have concerns on whether the imbalance problem under the SSL is an important research topic. I definitely agree with that the class imbalance problem is very important. However, it is weird when considering the class imbalance problem under the SSL scenario. As everyone knows, SSL is considered as pre-training under large scale unlabeled data (class-free), and then transfer to downstream tasks (not specific for the classification problem).
I believe the contributions are significant if this is the first paper to propose the imbalance problem under SSL setting. However, with several exists works as shown in the paper, I donot recognize the significance this paper contributes to the ML community. I think this paper will be stronger, if considering transfer to the class-imbalance object detection/semantic tasks.

**Questions:**

Address Weakness 2 and 3.

**Limitations:**

Yes

---

> ### Author Rebuttal · Authors · 2023-08-09
>
> We gratefully thank you for your time and efforts devoted to reviewing this paper and your constructive suggestions. Here are our detailed replies to your questions.
>
> > **Q1:** Cross-dataset transferring experiments.
>
> **A1:** We appreciate the reviewer’s constructive advice and conduct more comprehensive experiments on various cross-dataset transferring tasks to evaluate representation transferability as follows:
>
> - Pretraining datasets
>     - CC3M [1]. CC3M is a **large-scale, long-tailed** dataset with more than 3 million images. We pretrain SimCLR [11] and SimCLR with GH on CC3M for comparison.
> - Transferring to downstream classification
>     - Curated datasets: ImageNet [2] and Places [3]. We randomly subsample a balanced subset of ImageNet and Places for downstream finetuning, with the number of images per class set as 100.
>     - Fine-grained datasets: CUB200 [4], AirCraft [5], Stanford Cars [6], Stanford Dogs [7], NABirds [8]. These datasets requires semantic features to distinguish each category on the fine-grained granularity.
>     - We report the average accuracy of our CC3M-pretrained weights finetuning on classification on these datasets based on ResNet50 backbone.
> - Transferring to downstream object detection
>     - COCO2017 [9]. We report the bounding box AP of our CC3M-pretrained weights finetuning on object detection on COCO2017 using Faster-RCNN based on ResNet50-FPN backbone.
> - Transferring to downstream segmentation
>     - COCO2017 [9]. We report the mask AP of our CC3M-pretrained weights finetuning on segmentation on COCO2017 using Mask-RCNN based on ResNet50-FPN backbone.
>
> From the results in **Table R3-R4** in the complementary PDF file, we can see that our proposed GH consistently outperforms the baseline across various tasks and datasets. It further demonstrates the importance of **considering long-tailed data distribution under large-scale unlabeled data in the pretraining stage**. This can potentially be attributed to that our geometric harmonization motivates a more balanced and general emebdding space, improving the generalization ability of the pretrained model to a range of real-world downstream tasks.
>
> Besides, for more complete comparisons under other baselines, we will update the results in the submission once we finish all experiments.
>
> > **Q2:** Analysis of GH on other SSL methods.
>
> **A2:** Thanks for the constructive suggestions. We agree with the reviewer that our theorem and analyses are specific to contrastive learning. In terms of other non-contrastive SSL methods, we empirically show the superiority of our method on long-tailed data distribution in Section 4.4. Although it might not be straightforward to extend the theory to non-contrastive SSL methods, an explanation about the consistent superiority is that some non-contrastive methods still exhibit similar representation disparity with their contrastive counterpart, and our proposed method can similarly reallocate the geometric distribution to counteract the distorted embedding space. Specially, the recent study [10] theoretically and emprically explore the equivalence between contrastive and non-contrastive criterion, which may shed light on the intrinsic mechanism of how our GH benefits non-contrastive paradigm.
>
> We appreciate the reviewer’s question, and will include these discussions about the our theorem in the revision for clarity.
>
> > **Q3:** The importance of considering imbalancing learning in SSL scenarios, the contribution of GH and the transferring experiments on object detecion/segmentation tasks.
>
> **A3:** Thanks for the comments although it is a challenging point of view. Without offense, we might be persistent that considering imbalancing learning in SSL scenarios can also be critical to the representation generalization. Following the reviewer's advice in Q1, we conduct the representation learning on large-scale long-tailed CC3M and  verify that our GH inherently provides merits for the generalization on different downstream tasks, including detection and segmentation as suggested. As completely transferring all baselines in large-scale datasets is time-consuming and computationally expensive, we will update the comprehensive experiments in the submission as soon as possible.
>
> Besides, we would like to reclarify our taxonomy as described in **Lines 28-38** that all previous explorations target to improve self-supervised long-tailed learning, but from different aspects such as reweight/optimization techniques, architecture design and data augmentation. However, **none of these methods focus on the intrinsic limitation of the contrastive learning**. Concretely, we provide more detailed aspects in **Table G1** in the general response for clarity.
>
> Furthermore, to the best of our knowledge, we are the first to **explicitly point out the concept of representation disparity, which is the key drawback we have investigated**, in self-supervised long-tailed learning. We believe that the undesired disparity is the intrinsic limitation of contrastive loss to hurt the representation quality.

---

> > ### Comment · Reviewer_krnL · 2023-08-19
> > **Response to authors' rebuttal**
> >
> > I acknowledge the response of the authors and I have read the authors responses. My concerns on “representation transferability" are addressed by the authors' additonal experiments. Besides, the results of additional experiments pretrained on CC3M also increases my interest in the imbalance class under SSL setups. I raised my score from 5 to 6, and hope the authors can include the additional experiments in the revised version.

---

> > > ### Author Response · Authors · 2023-08-19
> > > **Thank you for the response**
> > >
> > > Dear Reviewer krnL,
> > >
> > > We sincerely appreciate you taking the time to review our responses and contributing to improve this paper. We will carefully follow reviewer's advice to incorporate all the addressed points with additional experiments in the updated version. Thank you once again for your dedicated and valuable contribution in reviewing our paper!
> > >
> > > Best,
> > >
> > > Authors of Paper 11308

---

### Author Rebuttal · Authors · 2023-08-09

We gratefully thank all the reviewers for their devoted efforts and constructive suggestions on this paper. We are glad that the reviewers have some positive impressions of our work, including:

- Explorations on an **important and useful** problem (M9kH, pT3P, G4d9).
- **The motivation is clear** (krnL, M9kH, TBJn).
- The method is **interesting, novel, solid, reasonable**, and can be **seamlessly** integrated into existing SSL methods (krnL, M9kH, TBJn, pT3P, G4d9, R4di).
- **Extensive and solid** experiments with **plenty of ablation studies** (krnL, M9kH, TBJn, pT3P).
- The paper is **well-written and easy to follow** (krnL, M9kH, TBJn, pT3P, G4d9).

We have addressed the reviewers' comments and concerns in **individual responses to each reviewer**. The reviews allowed us to improve our draft, and the changes made in our responses are summarized below:


- To back up the importance of the studies on self-supervised long-tailed learning, we provide further evidence with both the quantitative and qualitative empirical studies, see Table R3-R5, Figure R1.
- To address the concerns on the cross-dataset transferability of the proposed GH, we add more empirical evidence on various cross-dataset tasks pretrained on large-scale long-tailed CC3M, see Table R3-R4.
- We incorporate our method to more self-supervised methods, including BYOL and MoCo-v3 (See Table R1). Besides, we add empirical comparisons with several concurrent studies, including PMSN and TS (See Table R2).
- We further expand the ablation studies with comprehensive experiments, including sensitivity analysis of hyperparameters (See Figure R3), training configurations (See A2 to Reviewer pT3P), t-SNE visualization (See Figure R1) and uniformity analysis (See Table R5).
- We provide a more in-depth and comprehensive analysis on several points, including difference between SSL-LT and S-LT, the limitation of vanilla contrastive learning, the contribution of GH to SSL-LT, analysis of GH on other SSL methods, analysis of assumption on neighboring samples in the same class and advantages of surrogate label allocation.

**We appreciate all reviewers' great effort again!** We have tried our best to address your concerns and improve the paper following the suggestions. **Would you mind checking it and confirming if there are any unclear parts?**

**Tables:**

[**Table G1.** Taxonomy of self-supervised long-tailed methods.]
| Method | Aspect | Description         |
| ------ | ----------- | ------------------- |
| Focal [12]  | Sample Reweighting        | Hard example mining |
|     rwSAM [13]  |   Optimization Surface         |  Data-dependent sharpness-aware minimization                   |
|   SDCLR [14] | Model  Pruning    | Model pruning and self-contrast  |
| DnC [15]   | Model  Capacity   | Multi-expert ensemble            |
| BCL [16]   | Data  Augmentation  | Memorization-guided augmentation |
| GH     | Loss  Limitation      | Geometric harmonization          |

**References:**

[1] Piyush Sharma et al. Conceptual captions: A cleaned, hypernymed, image alt-text dataset for automatic image captioning. ACL 2018.

[2] Jia Deng et al. Imagenet: A large-scale hierarchical image database. CVPR 2009.

[3] Bolei Zhou et al. Places: A 10 million image database for scene recognition. TPAMI 2017.

[4] Catherine Wah et al. The caltech-ucsd birds-200-2011 dataset. 2011.

[5] Subhransu Maji et al. Fine-grained visual classification of aircraft. 2013.

[6] Jonathan Krause et al. 3d object representations for fine-grained categorization. ICCV workshop 2013.

[7] Aditya Khosla et al. Novel dataset for fine-grained image categorization: Stanford dogs. CVPR workshop 2011.

[8] Van Horn et al. Building a bird recognition app and large scale dataset with citizen scientists: The fine print in fine-grained dataset collection. CVPR 2015.

[9] Tsung-Yi Lin et al. Microsoft coco: Common objects in context. ECCV 2014.

[10] Quentin Garrido et al. On the duality between contrastive and non-contrastive self-supervised learning. ICLR 2023.

[11] Ting Chen et al. A simple framework for contrastive learning of visual representations. ICML 2020.

[12] Tsung-Yi Lin et al. Focal loss for dense object detection. ICCV 2017.

[13] Hong Liu et al. Self-supervised learning is more robust to dataset imbalance. ICLR 2022.

[14] Ziyu Jiang et al. Self-damaging contrastive learning. ICML 2021.

[15] Yonglong Tian et al. Divide and contrast: Self-supervised learning from uncurated data. ICCV 2021.

[16] Zhihan Zhou et al. Contrastive Learning with Boosted Memorization. ICML 2022.

[17] Asano Yuki Markus et al. Self-labelling via simultaneous clustering and representation learning. ICLR 2019.

[18] Mathilde Caron et al. Unsupervised learning of visual features by contrasting cluster assignments. NeurIPS 2020.

[19] Jiye Liang et al. The K-means-type algorithms versus imbalanced data distributions. TFS 2012.

[20] Mahmoud Assran et al. The hidden uniform cluster prior in self-supervised learning. ICLR 2023.

[21] Anna Kukleva et al. Temperature schedules for self-supervised contrastive methods on long-tail data. ICLR 2023.

[22] Kaiming He et al. Momentum contrast for unsupervised visual representation learning. CVPR 2020.

[23] Xinlei Chen and Kaiming He. Exploring simple siamese representation learning. CVPR 2021.

[24] Jure Zbontar et al. Barlow twins: Self-supervised learning via redundancy reduction. ICML 2021.

[25] Jean-Bastien Grill et al. Bootstrap your own latent-a new approach to self-supervised learning. NeurIPS 2020.

[26] Xinlei Chen et al. An Empirical Study of Training Self-Supervised Vision Transformers. ICCV 2021.

[27] Kaiming He et al. Masked autoencoders are scalable vision learners. CVPR 2022.

[28] Tongzhou Wang and Phillip Isola. Understanding contrastive representation learning through alignment and uniformity on the hypersphere. ICML 2020.

[29] Tianhong Li et al. Targeted supervised contrastive learning for long-tailed recognition. CVPR 2022.

---

> ### Comment · Area_Chair_MciK · 2023-08-18
>
> Thank the authors for the rebuttal. PCs and I have reminded the reviewers to respond to the rebuttals as soon as possible. The final decision will depend on both the reviews and rebuttal.
>
> @Reviewers: This message is yet another reminder. Please try to respond to the rebuttal asap.
>
> --AC

---

> > ### Author Response · Authors · 2023-08-19
> >
> > Dear Chairs,
> >
> > We sincerely appreciate your valuable time and effort in chairing our submission. We will carefully follow all reviewers' suggestions and discussions in this phase to improve this submission.
> >
> > Best,
> >
> > Authors of Paper 11308

---

### Decision · Program_Chairs · 2023-09-21

**Decision:**

Accept (spotlight)

**Comment:**

Six experts reviewed the paper, and all were positive after the rebuttal phase. The reviewers liked the idea and implementations, and the authors' experiments under the transfer setting are well appreciated. Hence, the decision is to recommend the paper for acceptance. We hope the authors will revise the paper according to the reviewers' feedback and the rebuttals.